

# Probabilistic analysis of ambiguities in radar echo direction of arrival from meteors

Daniel Kastinen[1,2] and Johan Kero[1]

[1]Swedish Institute of Space Physics (IRF), Box 812, SE-98128 Kiruna, Sweden
[2]Umeå University, Department of Physics, SE-90187 Umeå, Sweden

**Correspondence:** Daniel Kastinen (daniel.kastinen@irf.se)

**Abstract.** Meteors and hard targets produce coherent radar echoes. If measured with an interferometric radar system, these echoes can be used to determine the position of the target through finding the Direction Of Arrival (DOA) of the incoming echo onto the radar. If the DOA of meteor trail plasma drifting with the ambient atmosphere is determined, the neutral wind at the observation altitude can be calculated. Specular meteor trail radars have become widespread scientific instruments to study atmospheric dynamics. Meteor head echo measurements also contribute to studies of the atmosphere as the meteoroid input of extraterrestrial material is relevant for a plethora of atmospheric phenomena. Depending on the spatial configuration of radar receiving antennas and their individual gain patterns, there may be an ambiguity problem when determining the DOA of an echo. Radars that are theoretically ambiguity free are known to still have ambiguities that depend on the total radar Signal to Noise Ratio (SNR). In this study we investigate robust methods which are easy to implement to determine the effect of ambiguities on any hard target DOA determination by interferometric radar systems. We apply these methods specifically to simulate four different radar systems measuring meteor head and trail echoes using the multiple signal classification (MUSIC) DOA determination algorithm. The four radar systems are the middle and upper atmosphere (MU) radar in Japan, a generic Jones $2.5\lambda$ specular meteor trail radar configuration, the Middle Atmosphere Alomar Radar System (MAARSY) radar in Norway and the The Program of the Antarctic Syowa Mesosphere Stratosphere Troposphere Incoherent Scatter (PANSY) radar in the Antarctic. We also examined a slightly perturbed Jones $2.5\lambda$ configuration used as a meteor trail echo receiver for the PANSY radar. All the results are derived from simulations and their purpose is to grant understanding of the behaviour of DOA determination. General results are: there may be a region of SNRs where ambiguities are relevant; Monte Carlo simulation determines this region and if it exists; the MUSIC function peak value is directly correlated with the ambiguous region; a Bayesian method is presented that may be able to analyse echoes from this region; the DOA of echoes with SNRs larger then this region are perfectly determined; the DOA of echoes with SNRs smaller then this region completely fail to be determined; the location of this region is shifted based on the total SNR versus the channel SNR in the direction of the target; asymmetric subgroups can cause ambiguities even for "ambiguity free" radars. For a DOA located at the zenith, the end of the ambiguous region is located at 17 dB SNR for the MU radar and 3 dB SNR for the PANSY radar. The Jones radars are usually used to measure specular trail echoes far from zenith. The ambiguous region for a DOA at $75.5°$ elevation and $0°$ azimuth ends at 12 dB SNR. Using the Bayesian method it may be possible to analyse echoes down to 4 dB SNR for the Jones configuration, given enough data points from the same target. The PANSY meteor trail echo receiver did not deviate significantly from the





generic Jones configuration. The MAARSY radar could not resolve arbitrary DOAs sufficiently well to determine a stable region. However, if the DOA search is restricted to 70° elevation or above by assumption, stable DOA determination occurs above 15 dB SNR.

*Copyright statement.* TEXT

## 1 Introduction

Radar systems are a vital part of current research infrastructure. They are used for a wide variety of novel, (e.g Sato et al., 2014; Kero et al., 2012b; McCrea et al., 2015), and routine remote sensing observations (e.g Hocking, 2005, and references therein). One subset of these observations are objects and phenomena in the atmosphere that produce coherent radar echos.

Meteor head and trail echoes, satellite and space debris echos, polar mesospheric echos, field aligned irregularities and many more phenomena fall under this category. However, to discern the position and motion of these radar targets, interferometric or multi-static radar systems must be used.

When determining the position of an object by interferometry, there is an ambiguity problem (Schmidt, 1986). The position is determined by finding the Direction Of Arrival (DOA) of the incoming echo onto the radar. Depending on the spatial

configuration of the receiving antennas and their individual gain patterns, the voltage response can be the same for several different plane wave DOA's, thereby making it impossible to determine the correct direction. This problem is general to all DOA determinations made by radar systems. In this study it is put in the context of meteor head- and trail echo observations.

Every day the Earth's atmosphere is bombarded by billions of dust-sized particles and larger pieces of material from space. This incoming material gives us a unique opportunity to examine the motion and population of small bodies in the solar system

(e.g Vaubaillon et al., 2005a, b; Kastinen and Kero, 2017). Objects with sizes between 100 microns and 1 metre moving in interplanetary space are called meteoroids. Meteoroids originate from comets and asteroids, they are abundant and can have high velocities (Whipple, 1951). When meteoroids enter the atmosphere they burn up causing a phenomenon called a meteor (Ceplecha et al., 1998). The meteor itself can be divided into two parts that function as hard targets: the dense plasma co-moving with the ablating meteoroid and the trail of diffusing plasma left in the atmosphere. These generate the meteor head

and trail echoes.

Meteor trail plasma drifts with the ambient atmosphere. The drift velocity is therefore a measure of the neutral wind at the observation altitude. The typical ablation altitude were meteor phenomena occur lie between 70 and 130 km. This region is characterized by variability driven by atmospheric tides as well as planetary and smaller scale gravity waves. Specular meteor trail radars have become widespread scientific instruments to study atmospheric dynamics deployed at locations covering

latitudes from Antarctica to the Arctic Svalbard (Kero et al., 2019). To calculate the neutral wind, the DOA of the specular echo must be determined.

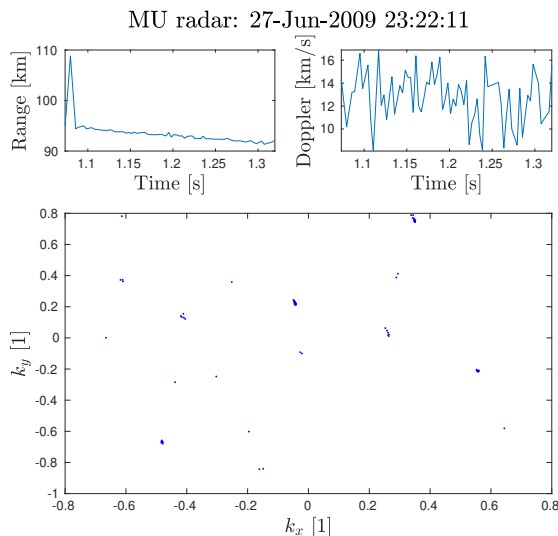

**Figure 1.** Example meteor head event measured with the MU radar which seem ambiguous. The upper left panel is the one way range, the upper right panel is the line of sight velocity, and the bottom panel is the normalized wave vector ground projection. The wave vector $+y$-axis is aligned with North and the $+x$-axis with the East. The SNR varied between 8-16 dB for this event. The goal of this paper is to understand enough about DOA determination behaviour to investigate these types of events.

Due to the altitude distribution of meteor phenomena, the far field approximation is almost always valid, which means that an incoming echo can be modeled as a plane wave (Kildal, 2015). The only exception is the Arecibo radio telescope due to its 300 m diameter large spherical reflector at a 430 MHz operating frequency. Using an interferometric radar system to determine
DOA of a meteor head echo as a function of time allows the construction of a meteoroid trajectory (e.g Kero et al., 2012b; Jones et al., 2005, 1998; Szasz et al., 2008; Chau and Woodman, 2004; Close et al., 2000). This trajectory is the base for computing the meteor position and meteoroid velocity, radar cross section, and for reconstructing the original meteoroid solar system orbit.

There are analytic methods to determine all the ambiguities present in a radar system, although these scale poorly and have
several restrictions (Kastinen, 2018; Schmidt, 1986). Systems that have no theoretical ambiguities also suffer from ambiguous DOA solutions due to noise (Kastinen, 2018; Jones et al., 1998). These so-called noise-induced ambiguities are not multiple solutions to the DOA determination. Instead, the DOA determination output becomes a stochastic variable that is no longer centered on the true DOA but is spread out between several different DOA solutions with similar radar responses. Thus, when determining the DOA of a noisy signal there is a probability of miss-classifying the DOA. This miss-classification probability
is separate from the DOA error introduced by the noise that is usually the focus in measurement pipelines (Kero et al., 2012b) and depends on the Signal to Noise Ratio (SNR) of the received signal.

The existence of head echo events such as the one illustrated in Fig. 1 is the reason that prompted this study. This meteor is seemingly jumping from place to place in the sky even though the range and line of sight velocity was well determined.



This was a special case among thousands of other successful and validated measurements using the same analysis and system. Hence, the goal of this study is to understand enough about DOA determination behaviour to investigate these types of events, especially today when analysis is automated and databases can contain millions of events (Campbell-Brown, 2019). If even a small fraction of these events stand out as interesting but are the result of ambiguities or other artifacts, consequent research can be negatively influenced. On the other hand, given strange results, large portions of data may be discarded when in fact some are not a consequence of ambiguities or algorithmic errors (Schult et al., 2013). Having a good understanding of DOA determination behaviour may also allow us to analyse events with lower SNR then currently (Jones et al., 1998).

There are no methods, to our knowledge, to resolve noise-induced ambiguities in DOA determinations or to determine the probability of miss-classification. We have therefore extended upon the study performed in Kastinen (2018).

In Sect. 2 we present a numerical method for determining the DOA ambiguities, which works irrespective of whether the ambiguities are noise-induced or not. The method can be applied on arbitrary radar systems and experience no large scaling problems with system complexity. It also allows for arbitrary receiver models to be used.

Section 3 provides an overview of how we have applied the the multiple signal classification (MUSIC) algorithm. The MUSIC method allows for an arbitrary sensor response model and can thus be applied on any radar system.

We have focused on radars measuring the meteor phenomena. However, the analysis methods applied are usable on any kind of interferometric hard-target detections made by radars. The radars that we have applied these techniques on are described in Sect. 4 and the results are presented in Sect. 5. The results for each system is also discussed in its respective subsection. Finally we conclude the results and discuss the overall results in Sect. 6.

## 2 Method

### 2.1 Ambiguities

To find the ambiguities present when determining the DOA, we need a radar sensor response model $\mathbf{\Phi}$. A model for a radar with antennas at locations $\mathbf{r}_i$, with individual complex gain functions $g_i(\mathbf{k})$, receiving a plane wave of amplitude $A$ is described by

$$\mathbf{\Phi}(\mathbf{k}) = \begin{pmatrix} A g_1(\mathbf{k}) e^{-i\langle \mathbf{k}, \mathbf{r}_1 \rangle_{\mathbb{R}^3}} \\ \vdots \\ A g_N(\mathbf{k}) e^{-i\langle \mathbf{k}, \mathbf{r}_N \rangle_{\mathbb{R}^3}} \end{pmatrix}. \tag{1}$$

Here $\mathbf{k}$ is the wave-vector of the incoming plane wave. We denote the inner product of a space $X$ by $\langle \cdot, \cdot \rangle_X$, i.e. $\langle \cdot, \cdot \rangle_{\mathbb{R}^3}$ is the real 3-dimensional inner product. In the case of radar systems with sub-arrays, the $g_i(\mathbf{k})$ functions can be defined as

$$g_j(\mathbf{k}) = \sum_{l=1}^{N_j} \gamma_{jl}(\mathbf{k}) e^{-i\langle \mathbf{k}, \mathbf{r}_j - \boldsymbol{\rho}_{jl} \rangle_{\mathbb{R}^3}}, \tag{2}$$





where $\gamma_{jl}(\mathbf{k})$ are the antennas' individual gain functions and $\boldsymbol{\rho}_{jl}$ are the sub-array antenna locations. In this case the $\mathbf{r}_i$ locations are the geometric centers of the sub-arrays, i.e. the phase centers. In all radar systems we consider, the same type of antennas are used throughout the system. As such, we can use a common function for all antennas $\gamma_{jl}(\mathbf{k}) = \gamma(\mathbf{k})$.

Usually, the wave amplitude $A$ is unknown and the DOA determination algorithm should therefore be invariant of signal amplitude $|\boldsymbol{\Phi}(\mathbf{k})|$. As such, our definition of an ambiguity should also be invariant of signal amplitude.

In the pursuit of an analytical solution, Kastinen (2018) was unable to include the variable gain patterns $g_i$ of the radar channels in the formula for finding ambiguities. The calculation method presented there scaled badly with the number of channels in the system and was not invariant to signal amplitude. We have resolved these issues, thereby allowing for any model $\boldsymbol{\Phi}$ to be used, by numerically finding ambiguities on a case by case basis.

Ambiguities are formed when

$$\frac{\boldsymbol{\Phi}(\mathbf{k}_0)}{|\boldsymbol{\Phi}(\mathbf{k}_0)|} \approx \frac{\boldsymbol{\Phi}(\mathbf{k})}{|\boldsymbol{\Phi}(\mathbf{k})|} : \mathbf{k}_0 \not\approx \mathbf{k}. \tag{3}$$

Exactly what the conditions "approximate to" and "not approximate to" mean in this definition needs to be decided on a case by case basis as explained further below. The normalized sensor response model is written as

$$\frac{\boldsymbol{\Phi}(\mathbf{k})}{|\boldsymbol{\Phi}(\mathbf{k})|} = \hat{\boldsymbol{\Phi}}(\mathbf{k}). \tag{4}$$

Equation 3 is invariant to the individual antenna gain $\gamma(\mathbf{k})$. Thus, we may define $\gamma(\mathbf{k}) = 1$ for all examined radar systems.

We call an ambiguity perfect, i.e. unambiguous DOA determination is impossible even at infinite SNR, if $\hat{\boldsymbol{\Phi}}(\mathbf{k}_0) = \hat{\boldsymbol{\Phi}}(\mathbf{k})$ : $\mathbf{k}_0 \neq \mathbf{k}$.

We define a set of ambiguities to $\mathbf{k}_0$ as the vectors $\mathbf{k}$ that fulfill Eq. 3, i.e.

$$\Omega(\mathbf{k}_0) = \{\mathbf{k} : \hat{\boldsymbol{\Phi}}(\mathbf{k}_0) \approx \hat{\boldsymbol{\Phi}}(\mathbf{k}) : \mathbf{k}_0 \not\approx \mathbf{k}\}. \tag{5}$$

It is important to note that it is not sufficient to calculate the set of ambiguities for only one $\mathbf{k}_0$, as this set may not display the same pattern as the set for another direction $\mathbf{k}_1$.

Following the definition in Eq. 3, an indicator function of ambiguities is the normalized sensor response distance

$$d(\mathbf{k}) = \left| \hat{\boldsymbol{\Phi}}(\mathbf{k}_0) - \hat{\boldsymbol{\Phi}}(\mathbf{k}) \right|. \tag{6}$$

If the set $\Omega(\mathbf{k}_0)$ is finite, the distance function $d$ must have valleys with a single point bottom at every $\mathbf{k}$ in $\Omega(\mathbf{k}_0)$. These valleys $\mathbf{k}$ are separated from $\mathbf{k}_0$ and have a depth $d$ that are used to decide if they are included in the ambiguity set $\Omega(\mathbf{k}_0)$ or not. As the valleys identify ambiguities, we have implemented a scattered gradient descent method to determine the ambiguity set $\Omega$ for a given $\mathbf{k}_0$. The step by step method is as follows:

1. Define a source wave direction $\mathbf{k}_0$

2. Generate a set of $n$ start points $\{\mathbf{a}_i\}$ distributed (e.g uniformly) on the hemisphere

3. For each start point, do a gradient descent search using the gradient of Eq. 6, $\nabla d(\mathbf{k})$.



4. Collect the valley locations $\{\mathbf{b}_i\}$ and valley depths $\{d(\mathbf{b}_i)\}$

5. Remove duplicate results yielding the set of all ambiguities and their depths $\{\mathbf{k}_j, d(\mathbf{k}_j)\}$.

This method can be used with any sensor response model to find ambiguities. After the set $\{\mathbf{k}_j, d(\mathbf{k}_j)\}$ is acquired, it is necessary to filter the set based on what is deemed approximate and not approximate as per Eq. 5. This is done based on some maximum valley depth $\epsilon_d$ and some minimum separation from $\mathbf{k}_0$. The filtering prevents the inclusion of ambiguities that only appear at unrealistically low SNR's. We will from here on denote this filtered set of ambiguities by $\Omega(\mathbf{k}_0) = \{\mathbf{k}_j\}$.

An important factor to note is that these ambiguities are **not** necessarily transitive relations. They are only transitive when the distance is 0, i.e. they occupy the same point in sensor response space. The non-transitive relationship means that for $d > 0$, if $\mathbf{k}_1$ is ambiguous with $\mathbf{k}_2$ and $\mathbf{k}_2$ is ambiguous with $\mathbf{k}_3$, $\mathbf{k}_1$ does not have to be ambiguous with $\mathbf{k}_3$.

## 2.2 Noise

All the radar systems considered in this study have operating frequencies in the Very High Frequency (30–300 MHz) range. In this range, the galactic background radiation dominates the noise (e.g Bianchi and Meloni, 2007). This noise can be well modeled (e.g Polisensky, 2007). When measured by an antenna, the noise is modeled as a circularly-symmetric complex normal random variable. Such a distribution is defined as $\mathcal{CN}(\boldsymbol{\mu} = \mathbf{0}, \Sigma, C = 0)$ where $\mu$ is the complex mean vector, $\Sigma$ is the covariance matrix and $C$ is the relation matrix. We assume that the noise dynamics is the same for every channel of a $N$ channel radar system. We can thus use $N$ random variables in a one-dimensional complex space instead of an $N$ dimensional complex space. Furthermore, since the distribution is circularly-symmetric we define the controlling variable to be the variance of a single component, i.e. the real or imaginary variance $\sigma_c^2$. The covariance matrix then becomes $\Sigma = 2\sigma_c^2$. The sensor noise is defined as

$$\boldsymbol{\xi} = \begin{pmatrix} \xi_1 \\ \vdots \\ \xi_N \end{pmatrix}, \tag{7}$$

$$\xi_i \sim \mathcal{CN}(0, 2\sigma_c^2, 0). \tag{8}$$

In pseudo code the noise can now be simulated as `xi = (rand_normal(N) + i*rand_normal(N))*sigma_c`.

## 2.3 Signal to Noise ratio

In order to relate results from simulations to measured data, the noise-controlling variable $\sigma_c$ needs to be related to a measured SNR. We have chosen to use an SNR that is calculated after coherently integrating over all radar channels. The noisy signal power is then defined as

$$P = \left| \sum_{i=1}^N \Psi_i + \xi_i \right|^2. \tag{9}$$





If we propagate the stochastic variables using standard properties of complex normal distributions we find that

$$N\sigma_c^2 P \sim \chi^2(\lambda, 2),$$ (10)

where $\chi^2(\lambda, 2)$ is the non-central chi-squared distribution of order 2 with $\lambda$ parameter

$$\lambda = \frac{1}{N}\left(\frac{AG(\mathbf{k})}{\sigma_c}\right)^2.$$ (11)

The order of a non-central chi-squared distribution is equal to the number of squared normal distributions that are summed, while the lambda parameter is related to their mean values. Here $A$ is the signal amplitude and $G(\mathbf{k})$ describes the one-directional gain in the source direction, i.e

$$G(\mathbf{k}) = \left|\sum_{i=1}^{N}\frac{1}{A}\Psi_i(\mathbf{k})\right|.$$ (12)

The expected value of the power is then

$$\mathbb{E}[P] = (AG(\mathbf{k}))^2 + 2N\sigma_c^2.$$ (13)

Setting $A = 0$ gives the noise power $\mathbb{E}[P_n] = 2N\sigma_c^2$ and setting $\sigma_c = 0$ gives the signal power $\mathbb{E}[P_s] = (AG(\mathbf{k}))^2$. SNR is defined as the ratio between the the signal power and the noise power, i.e.

$$\text{SNR} = \frac{\mathbb{E}[P_s]}{\mathbb{E}[P_n]} = \frac{\mathbb{E}[P]}{\mathbb{E}[P_n]} - 1 = \frac{1}{2N}\left(\frac{AG(\mathbf{k})}{\sigma_c}\right)^2.$$ (14)

Assuming we have two measurements, one of the noise power $\mathbb{E}[P_n]$ and one of the noisy signal power $\mathbb{E}[P]$, an SNR that is equivalent to that used in our simulations can be calculated for any detected signal. Using Eq. 14, an appropriate $\sigma_c$ for a given SNR can be chosen for a simulation.

## 2.4 Direct Monte Carlo

Given a sensor response model and a noise model we can perform a direct Monte Carlo (MC) on any DOA determination algorithm. Given a true direction $\mathbf{k}_j$, the theoretical noisy sensor response model is $\mathbf{\Psi}(\mathbf{k}_j) + \boldsymbol{\xi}$. Then, a DOA determination algorithm $F$ can find an estimation of the source direction as,

$$F(\mathbf{\Psi}(\mathbf{k}_j) + \boldsymbol{\xi}) = \tilde{\mathbf{k}}.$$ (15)

Thus, the estimated source direction $\tilde{\mathbf{k}}$ also becomes a distribution. We can sample this DOA determination output distribution by sampling the noisy signal distribution $\mathbf{\Psi}(\mathbf{k}_j) + \boldsymbol{\xi}$ and applying the DOA determination algorithm $F$ on each sample. An example MC sampling of such a DOA output distribution is illustrated in Fig. 2. This example was generated using the generic Jones $2.5\lambda$ sensor response model further described in Sect. 4.1. This radar model does not contain any perfect ambiguities,





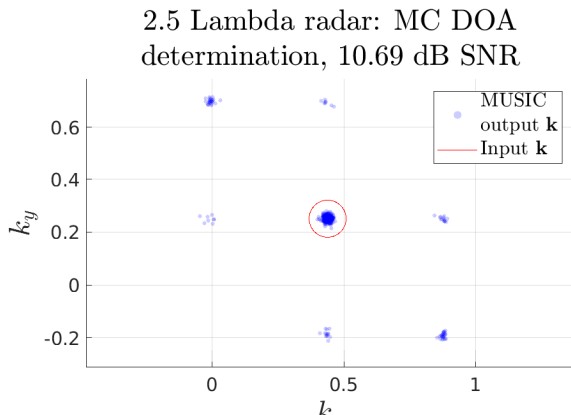

**Figure 2.** Example MC DOA determination simulation with 500 samples. The generic Jones $2.5\lambda$ radar model as described by Eq. 29 was used to simulate the raw data. Noise was introduced to an equivalent SNR of 10 dB. The simulated noisy raw data was analysed using the two-step MUSIC algorithm (Sect. 3). The Input DOA was located at $0°$ azimuth and $75.5°$ elevation. At this SNR, noise-induced ambiguities are clearly visible. The probability that the output is associated with the input is 79%.

yet at this SNR the DOA output is scholastically clustered around noise-induced ambiguities. The DOA determination was made using the multiple signal classification (MUSIC) algorithm described in Sect. 3. The interesting aspect that will allow qualitative evaluation of measurement data is how the DOA output behaviour evolves as a function of SNR, true DOA, sensor response model, and DOA determination algorithm. In Sect 5 we examine the first three of these components while keeping the DOA determination algorithm fixed.

## 2.5 Discretising the problem

The example MC DOA determination simulation in Fig. 2 contain apparent noise-induced ambiguities alongside the spread of the DOA estimation around the input direction and its ambiguities. This sampling of $\tilde{\mathbf{k}}$ represents a continuous distribution that contains information about both the DOA determination accuracy and possible ambiguities. There are many works that describe the error distribution of MUSIC for DOA determination in radars (e.g Kangas et al., 1994, 1996; Ferreol et al., 2006). However, we focus on the probability of ambiguous DOA output and general algorithm stability. Therefore we discretise the problem by using the set of known ambiguities $\mathbf{k}_i \in \Omega(\mathbf{k}_0)$ described in Sect. 2.1. To account for a limited DOA determination accuracy, we choose an inclusion distance $s$ in the wave vector ground projection plane. This distance determines the region around all ambiguities, as well as the true direction, within which we consider that particular ambiguity "chosen" by the algorithm. Thus, in the discretisation process, outputs will be considered as associated to either an ambiguity or the true DOA, otherwise they have no association.





Practically, if

$$(k_{ix} - \tilde{k}_x)^2 + (k_{iy} - \tilde{k}_y)^2 < s^2, \tag{16}$$

then the sample $\tilde{\mathbf{k}}$ is counted towards the probability $P_i$ for output point $\mathbf{k}_i$ assuming true input $\mathbf{k}_0$. The samples which cannot be associated with any inclusion region are considered as algorithm failures, i.e. $P_f = 1 - \sum_i P_i$.

We are interested in the misclassification and algorithm failure probability. We examine this by regarding the source as a variable $j$ and constructing a discrete probability $P_{ij}$ as a function of both source and output location.

Even though ambiguities are not necessarily transitive relations, as mentioned in Sect. 2.1, there may still be some overlap in the inclusion regions when we define them as $\mathbf{k}_i \in \Omega(\mathbf{k}_j)$. We therefore need to spend some thought on how to practically discretise the problem. If we consider the usage of the simulations as a tool for evaluating observations, it can be assumed that we have an observed $\mathbf{k}_0$. We then know that either $\mathbf{k}_0$ or one of its ambiguities $\Omega(\mathbf{k}_0)$ is the source. Consider that an ambiguity $\mathbf{k}_i \in \Omega(\mathbf{k}_0)$ would have an ambiguity $\mathbf{k}_h \in \Omega(\mathbf{k}_i)$ which is not part of our original set $\mathbf{k}_h \notin \Omega(\mathbf{k}_0)$. The probability that $\mathbf{k}_h$ is the source is zero as it could not have generated our observation $\mathbf{k}_0$. Following this line of reasoning, discretisation can be based on the members of the set $\Omega(\mathbf{k}_0)$ only. However, this poses a problem when a set of multiple observations from the same radar target are available that are not gathered around one point: which of the observed clusters should be used as $\mathbf{k}_0$?

To circumvent this problem, we form two separate sets of points $\Omega_X$ and $\Omega_Y$. $\Omega_X$ is the set of points used as sources $j$ in the simulation, and the set $\Omega_Y$ is used as output regions $i$. The set $\Omega_X$ is the ambiguity set of any of the observed points, $\Omega_X = \Omega(\mathbf{k}_0)$. We know that given an observation, the true source has to be contained in this set. Assuming all observations are from the same event, the choice of $\mathbf{k}_0$ does not matter. The set $\Omega_Y$ is then chosen as the collected set of all ambiguities to the simulated sources, i.e. $\Omega_Y = \bigcup_j \Omega(\mathbf{k}_j) : \mathbf{k}_j \in \Omega_X$. This approach is more tailored towards analysing observations rather then just classifying the ambiguity situation given a certain true source. If there are no individual ambiguity sets that includes all groups of measured output points, other effects are influencing the DOA determination. These effects could be radar phase calibration issues (e.g. Chau et al., 2014), antenna malfunctions or erroneous phenomena models (e.g. multiple simultaneous signals, signal interference, wave diffraction).

There are some practical consideration when implementing the construction of $\Omega_X$ and $\Omega_Y$: duplicate locations should not be included in $\Omega_Y$. These are handled by removal based on closeness in relation to the inclusion radius. The ordering of the sets $\Omega_X$ and $\Omega_Y$ are so that the first elements of $\Omega_Y$ correspond to the elements of $\Omega_X$ for clarity when examining simulation results.

Using the definitions for $\Omega_X$ and $\Omega_Y$ we can represent the probability $P_{ij}$ as a matrix, excluding all non-relevant association probabilities. Taking a second look at Fig. 2, we can imagine how a column of this matrix would be constructed. For each DOA in $\Omega_Y$ we count the outputs that are inside its inclusion radius. This number divided by the total number of samples is the probability $P_{ij}$. For the particular example in Fig. 2 the probability that the output is associated with the true input is 79%.

The columns of the matrix $P_{ij}$ describe different DOA inputs for the simulation and its rows describe the probability that the DOA determination algorithm outputs a location as the result. The most desirable form of this matrix would be a diagonal unit





matrix, i.e., given true input $j$ the DOA algorithm always finds the corresponding location as the output. Unfortunately, this is not always the case as this probability matrix is a function of the SNR, $P_{ij}(\mathrm{SNR})$.

## 2.6 Bayesian inference

It is generally not advisable to use data that is ambiguous. To quantitatively describe what is "too ambiguous" for further usage is one of the goals of this study. For example, as illustrated in Fig. 2, which represents a simulation of a measured event, the event could be analysed given that enough independent echoes were measured. The reason why the event is usable is that the simulation shows that the true direction has by far the highest output probability. This means that one can pick the largest "cluster" of output DOAs and conclude that this probably represents the true DOA. However, for more complex radar systems

and other input DOAs, the situation would look different. This line of reasoning also does not give us a quantitative confidence in our choice of true DOA.

If there is a need to analyse ambiguous data we suggest a Bayesian approach. As an example, let us return again to the simulation presented in Fig. 2. We argued using the simulation results that given enough independent measurements of such an event, the measurements could be used to infer the true DOA of the target. Bayesian inference generalizes this type of

245 argumentation by optimally using all available information to assign probabilities to all possible true input DOAs.

Given a model with parameters $\mathbf{x}$ that represents an event which has generated some observations $D$, Bayesian inference can be used to find the probability distribution of possible model parameters. This distribution is called the posterior $\mathcal{P}(\mathbf{x})$. The posterior also includes a prior probability $\theta(\mathbf{x})$, i.e. what we think the distribution is before any observations. The observed data is used to update the prior distribution by use of a likelihood function $L$. This likelihood function determines how probable

the observed data is given the model parameters $\mathbf{x}$. The relationship between the prior $\theta$, likelihood $L$ and posterior $\mathcal{P}$ is given by Bayes' theorem,

$$\mathcal{P}(\mathbf{x}) = \frac{L(D|\mathbf{x})\theta(\mathbf{x})}{\int L(D|\mathbf{x})\theta(\mathbf{x})\mathrm{d}\mathbf{x}}. \tag{17}$$

Here $|$ indicates conditional probability, i.e. $L(D|\mathbf{x})$ is read as "the likelihood of observing $D$ given the parameters $\mathbf{x}$".

This approach is compatible with the problem at hand. Assuming the matrix of probabilities $P_{ij}$ is calculated and known,

the probability of observing $\mathbf{k}_i$ given the input $\mathbf{k}_j$ is exactly given by $P_{ij}$. Therefore,

$$L(\mathbf{k}_i|\mathbf{k}_j) = P_{ij}. \tag{18}$$

For the first observed DOA, the prior is uniform over all $\mathbf{k}_j$, i.e.

$$\theta(\mathbf{k}_j) = \frac{1}{N_I}, \tag{19}$$





where $N_I$ is the number of columns in $P_{ij}$, i..e the size of $\Omega_X$. If we observe $i = a_1$ as the first calculated DOA we can find the posterior as

$$
\mathcal{P}_1(\mathbf{k}_j) = \frac{P_{a_1 j}}{\sum\limits_{j=1}^{N_I} P_{a_1 j}}.
\tag{20}
$$

If subsequent DOA's are observed, we update the probability distribution over the model parameters by setting the last posterior as our prior and applying the same formula,

$$
\mathcal{P}_x(\mathbf{k}_j) = \frac{P_{a_x j} \mathcal{P}_{x-1}(\mathbf{k}_j)}{\sum\limits_{j=1}^{N_I} P_{a_x j} \mathcal{P}_{x-1}(\mathbf{k}_j)}.
\tag{21}
$$

If the observations $a_x$ are done at different SNRs, the $P_{ij}$ matrix should be allowed to change for every update $x$ as $P_{ij}(\mathrm{SNR}_x)$.

This method may be able to infer the true direction even in very ambiguous data. At the very least, this method provides a probability distribution over the possible directions, as exemplified in Sect 5.

## 2.7 Ambiguous measurement simulation

A method to investigate whether the Bayesian approach makes a significant improvement on analysis is measurement simulation. The matrix $P_{ij}$ describes the probability of the DOA determination generating the output indexed $i$ given a noisy measurement of the true input DOA indexed $j$. We can thus simulate measurements as a multinomial distribution where the distribution probabilities are given by a column $j$ of the $P_{ij}$ matrix. Sampling this multinomial distribution will give a set of $n$ simulated measurements as a list of location indices (i.e. output DOAs). Given this list of indices, Eq. 21 can be applied to calculate the most probable input given the simulated data without running the DOA determination algorithm. We can repeat this process to get a simulated distribution for the posterior $\mathcal{P}_x(\mathbf{k}_j)$. This distribution is what we would expect to see as inference results from a measurement series of $n$ points given some true input. This distribution is useful for evaluating both the method itself and the radar system as it contains the probability of Bayesian inference finding the true input. This information can thus also provide the minimum SNR and measurement number needed to achieve a desired success rate in DOA determination, assuming the $P_{ij}$ matrices accurately model reality.

## 3 DOA determination

For the purpose of consistency and simplicity we have used the same DOA determination method for all radar systems examined: the MUSIC algorithm (Schmidt, 1986). This method allows for an arbitrary sensor response model $\mathbf{\Psi}$ and can thus be applied on all systems. MUSIC is practically equivalent to beam forming DOA methods but with reduced variance due to the subspace approach. We here give a short overview of how we have applied the MUSIC method.

We define a measured sensor response as the complex vector $\mathbf{x} \in \mathbb{C}^N$. The sensor response model in Eq. 1 refers to a so called decoded signal. The decoded signal is the signal coherently integrated over all temporal samples of a radar pulse. However,





the lowest level of raw data also contains these temporal samples of the radar pulse. Given $M$ temporal samples of the coded pulse, the measurement matrix then consists of of N rows and M columns as

$$X = \begin{pmatrix} \mathbf{x}_1 & \mathbf{x}_2 & \dots & \mathbf{x}_M \end{pmatrix}. \tag{22}$$

The correlation matrix $R$ of our measurements is calculated using matrix algebra as

$$R = \frac{1}{M} X X^\dagger. \tag{23}$$

The correlation matrix consists of coherently integrated channel-to-channel phase differences over the temporal samples. The eigenvalues of the correlation matrix correspond to signal powers and the eigenvectors corresponding to the largest eigenvalues span the signal subspace (Schmidt, 1986). If there is noise, the eigenspace spans the entire sensor configuration space, otherwise

it only spans the signal subspace. First, we extract the eigenvectors $\mathbf{P}_i$ and eigenvalues $\lambda_i$ of the correlation matrix using standard linear algebra methods. Then, assuming one signal subspace dimension, i.e. one signal from one direction, we define the noise subspace as the column space of

$$Q = \begin{pmatrix} \dots & \mathbf{P}_{j-1} & \mathbf{P}_{j+1} & \dots \end{pmatrix}, \tag{24}$$

where $\mathbf{P}_j$ corresponds to the largest eigenvalue $\lambda_j = \max(\{\lambda_i : i \in [1, M]\})$. This eigenvector represents the signal subspace.

MUSIC is a multiple signal classification method. If there are multiple signals present in the data, the second eigenvector and eigenvalue is associated with the second strongest signal, etc. As the column vectors of $Q$ form an orthonormal basis, consider the space

$$\mathbb{Q} = \mathrm{span}\left\{\mathbf{P}_1, \dots, \mathbf{P}_{j-1}, \mathbf{P}_{j+1}, \dots, \mathbf{P}_M\right\}. \tag{25}$$

The scalar projection function $P$ into this linear subspace $\mathbb{Q}$ is

$$P_\mathbb{Q}(\mathbf{x})^2 = \sum_{i=1, i \neq j}^{M} \langle \mathbf{P}_i, \mathbf{x} \rangle_{\mathbb{C}^M}^2 = |Q^\dagger \mathbf{x}|^2. \tag{26}$$

The space $\mathbb{Q}$ represents the noise, thus any space orthogonal to $\mathbb{Q}$ is a signal. The projection of a vector onto an orthogonal space is zero, thus we are searching for vectors $\mathbf{x}$ that minimizes $P_\mathbb{Q}(\mathbf{x})$.

We write the projection function in terms of matrix operations as

$$P_\mathbb{Q}(\mathbf{x})^2 = |Q^\dagger \mathbf{x}|^2 = (Q^\dagger \mathbf{x})^\dagger (Q^\dagger \mathbf{x}) = \mathbf{x}^\dagger Q Q^\dagger \mathbf{x}. \tag{27}$$

Normalizing the projection with respect to the input vector norm and inverting, we maximise instead of minimize, and find the familiar MUSIC function. As we have a model for $\mathbf{x}$ as a function of DOA, we set $\mathbf{x} = \mathbf{\Psi}(\mathbf{k})$ and find

$$f(\mathbf{k}) = \left( \frac{P_\mathbb{Q}(\mathbf{\Psi}(\mathbf{k}))^2}{|\mathbf{\Psi}(\mathbf{k})|^2} \right)^{-1} = \frac{\mathbf{\Psi}(\mathbf{k})^\dagger \mathbf{\Psi}(\mathbf{k})}{\mathbf{\Psi}(\mathbf{k})^\dagger Q Q^\dagger \mathbf{\Psi}(\mathbf{k})}, \tag{28}$$





which is the form usually recited in literature. This function needs to be maximized by an appropriate method to find the sensor response $\boldsymbol{\Psi}(\mathbf{k})$ that best matches the detected signal, thereby also determining the DOA, $\mathbf{k}$, of the signal.

We have chosen to apply a two-step maximization method. First, a finite grid search over all possible $\mathbf{k}$ was applied. Then, the maximum found during this grid search was used as an initial condition for a gradient ascent applied on $\nabla f(\mathbf{k})$ to find the peak point. Finally, the peak value is used as output, i.e. as the determined DOA of the signal.

However, there is no guarantee that the initial grid search will always be able to identify the correct slope as an initial condition for the gradient ascent. If the peak width is smaller then the grid size any slope may be found instead. To solve this 320 problem we also implemented an option of running multiple gradient ascents in parallel. When this option is enabled, instead of using only the maximum point from the grid search as a start value, the $N$ largest values that are separated from each other by at least $\delta X$ in $k_x, k_y$ space are used. The separation condition ensures that no two start points are located on the same slope. These $N$ start points are explored by a gradient ascent and the largest peak among them is chosen as the algorithm output.

## 4 Radar systems

The sensor response for all radars covered in this study were modeled using two different models, a simplified model

$$\boldsymbol{\Phi}(\mathbf{k}) = \begin{pmatrix} An_1 e^{-i\langle\mathbf{k},\mathbf{r}_1\rangle_{\mathbb{R}^3}} \\ \vdots \\ An_N e^{-i\langle\mathbf{k},\mathbf{r}_N\rangle_{\mathbb{R}^3}} \end{pmatrix}, \tag{29}$$

where $n_i$ is the number of antennas summed to that radar channel. And a model using the subgroup gain patterns

$$\boldsymbol{\Phi}(\mathbf{k}) = \begin{pmatrix} Ag_1(\mathbf{k}) e^{-i\langle\mathbf{k},\mathbf{r}_1\rangle_{\mathbb{R}^3}} \\ \vdots \\ Ag_N(\mathbf{k}) e^{-i\langle\mathbf{k},\mathbf{r}_N\rangle_{\mathbb{R}^3}} \end{pmatrix}, \tag{30}$$

$$g_j(\mathbf{k}) = \sum_{l=1}^{N_j} e^{-i\langle\mathbf{k},\mathbf{r}_j-\boldsymbol{\rho}_{jl}\rangle_{\mathbb{R}^3}}. \tag{31}$$

The exceptions are the Jones type radar systems where there are no subgroups but only single antennas. As previously mentioned, in these models $\mathbf{r}_i$ indicate the locations of individual antennas or the geometric centers of the sub-arrays, i.e. the phase centers.

In this study we have assumed that the antennas have omnidirectional gain. This is of course not the case, as mentioned in Sect. 2, but this assumption has no impact on the current study. As all radar systems examined have the same antennas 335 throughout the system, the individual gain function for an antenna cancels in any algorithm that is invariant to signal amplitude. However, in the implementation of a data analysis pipeline it is important to implement the individual antenna gain pattern $\gamma$ and the subgroup generated gain patterns in the sensor response model to be able to determine the radar cross section correctly.



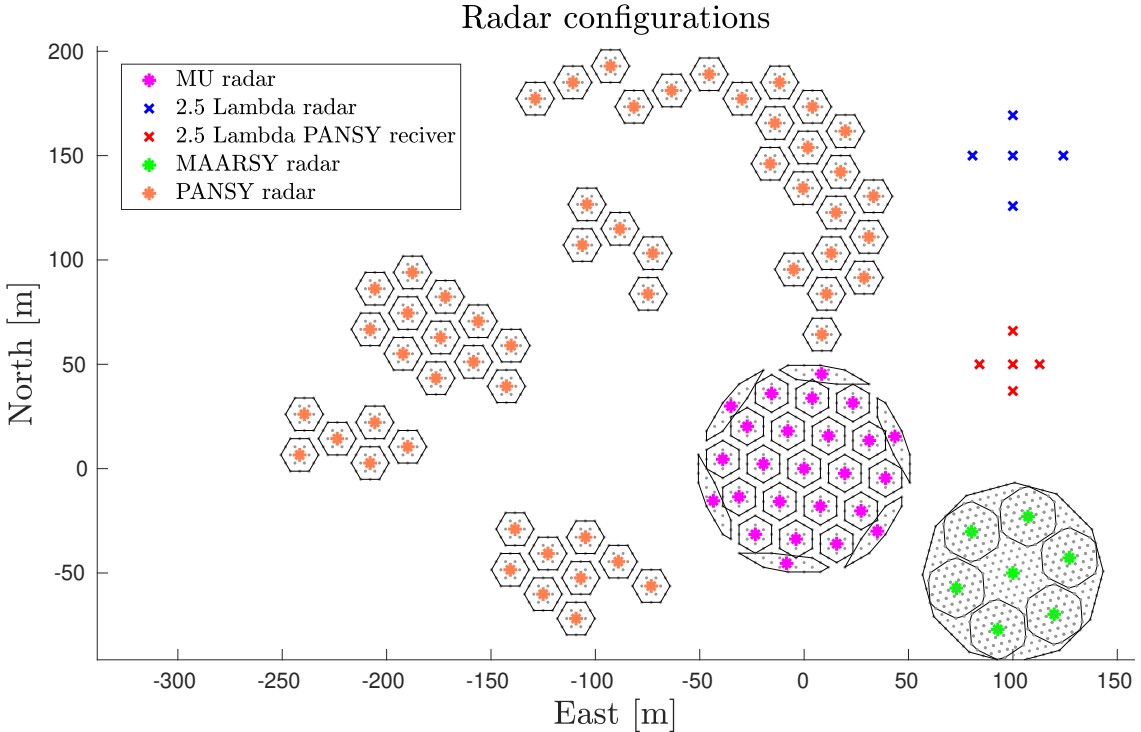

**Figure 3.** The different radars considered in the DOA determination study. For the radars which consists of subgroups of antennas the subgroup centers are colored radar-specific while antennas and subgroup borders are always grey and black. The Jones $2.5\lambda$ radars are single antenna channel radars so here colored markers indicate antennas.

We hereafter refer to the model in Eq. 29 as the **phase center model** and the model in Eq. 30 as the **subgroup model**.

In Fig. 3 the antenna positions of all examined radars are illustrated so that their individual configurations and sizes can be
compared.

### 4.1  Jones $2.5\lambda$ radar

Radar systems designed for studying meteor trail echoes commonly consist of a wide angle (all-sky) transmitter system and an interferometric receiver system (e.g. Jones et al., 1998; Hocking et al., 2001).

The receiver system design is beset by two problems (Jones et al., 1998): that antenna spaced more than $\lambda/2$ apart give rise
to ambiguities in the DOA, and that antennas spaced less than $\lambda/2$ apart give rise to strong mutual impedance. The so-called Jones $2.5\lambda$ radar configuration is an elegant solution suggested by Jones et al. (1998) as a remedy to the situation. The solution consists of using five antennas, one central antenna and two spaced by $2.5\lambda$ and $2.0\lambda$ in each of the two perpendicular cardinal directions (cf. Fig. 3).



As described by Jones et al. (1998), the phase measurements at the outer antennas relative to the central antenna can ideally be used to calculate an unambiguous determination of the echo DOA, taking advantage of the fact that the internal antenna distances to the central antenna differs by $\lambda/2$. Furthermore, the phase difference of antennas with $4.5\lambda$ spacing is used to give better angular precision at the cost of ambiguous DOA, the most probable solution of which is then selected using the $\lambda/2$ phase difference.

Holdsworth (2005) investigated the Jones antenna configuration and found that the usage of 2.5, 3 and $5.5\lambda$ spacings could produce more accurate echo DOA. Younger and Reid (2017) developed the concept further and presented a solution which utilise all possible antenna pairs of a meteor radar antenna configuration, similarly to the DOA calculations using MUSIC in this paper. In addition to providing results in excellent agreement with the original interfermetric algorithm by Jones et al. (1998), the method presented by Younger and Reid (2017) as well as the MUSIC algorithm allows for different layouts.

The Jones antenna configuration has remained predominant in meteor radar installations and is often referred to as removing (in principle) any angular ambiguities (Hocking et al., 2001). However, as was pointed out already by Jones et al. (1998), the determination is sensitive to noise and only unambiguous if the SNR is large enough. The original simulations by Jones et al. (1998) showed that the method started to produce incorrect apparent echo directions for elevations greater than $30°$ when the SNR was below 17 dB, but that at the same time the fraction of these was small down to about 10 dB. The standardized SKiYMET software meteor detection data contains an ambiguity level classification. If the ambiguity parameter is equal to 1, the data was determined to be unambiguous, and if it is greater than 1 there is a possibility that the meteor was wrongly located (Hocking et al., 2001).

To our knowledge, there are no further quantitative investigations of the Jones $2.5\lambda$ radar configuration performance except for the studies mentioned above and references therein. The results of applying the method presented in this paper on the Jones $2.5\lambda$ radar to quantify noise-induced ambiguities are given in section 5.1. As the mentioned studies on ambiguities already exists, simulating a Jones $2.5\lambda$ radar also provides a good reference simulation for validation of the methods presented in Sect. 2.

## 4.2 MU radar

The 46.5 MHz Middle and Upper atmosphere (MU) radar near Shigaraki, Japan (34.85°N, 136.10°E) has a nominal peak transmitter power of 1 MW and a maximum beam duty cycle of 5%. The present setup of the MU radar hardware comprises a 25 channel digital receiver system. It was upgraded from the original setup (Fukao et al., 1985) in 2004 and is described by Hassenpflug et al. (2008). After the upgrade, the MU radar always transmit right-handed circular polarization and receive left-handed circular polarization, with a phase accuracy of $2°$. The output of each digital channel is the sum of the received radio signal from a subgroup of 19 Yagi antennas. The whole array consists of 475 antennas, evenly distributed in a 103 m circular aperture, with a main lobe maximum gain of 34 dB and a minimum half power beam width of $3.6°$. A schematic view of the array and the subgroups is given in Fig. 3.





Early meteor head echo measurements using the original setup with four receiver channels (Nishimura et al., 2001) are not investigated further in this study. The focus is instead on the current 25-channel setup, which has been used more extensively for hard targets such as meteors (e.g. Kero et al., 2011, 2012a, b, 2013; Fujiwara et al., 2016; Kastinen and Kero, 2017).

### 4.3 MAARSY radar

The new Middle Atmosphere Alomar Radar System (MAARSY) was constructed in 2009/2010 on the Norwegian island Andøya (69.30°N, 16.04°E) following similar design principles as the MU radar. It is a monostatic radar operated at 53.5 MHz with an active phased array antenna consisting of 433 Yagi antennas (Latteck et al., 2010). The antennas are, similarly to the MU radar, arranged in an equilateral triangle grid with $0.7\lambda$ (4 m) spacing, forming a 90 m circular aperture. This results in a rather symmetric radar beam with a maximum directive gain of 33.5 dB and a minimum half power beam width of 3.6°. Each
individual antenna is connected to a transceiver with independent phase control and output power up to 2 kW, enabling flexible beam forming, beam steering and approximately 800 kW peak transmitter power with 5% duty cycle.

The smallest MAARSY subarray unit consists of seven antennas distributed in a hexagonal pattern as illustrated in Fig. 4. The receiver system currently allows for 16 separate channels. Early meteor head echo observations with MAARSY used eight channels which were defined according to Fig. 3, where seven of the channels consisted of the combined input from seven
subarrays (i.e. 49 antennas) and the eighth channel contained the combined input from all antennas (Schult et al., 2013). Later meteor head echo observations have made use of the alternative MAARSY configuration vizualised in Fig. 4 (Schult et al., 2017).

The radiation pattern of MAARSY have been studied and validated through observations of cosmic radio sources (Renkwitz et al., 2012, 2013), scattering of a sounding rocket's payload (Renkwitz et al., 2015) and meteor head echoes (Renkwitz et al.,
2017). Methods have also been developed to calibrate and validate the measured phases of the individual channels using cosmic radio noise and meteor head echoes (Chau et al., 2014).

### 4.4 PANSY radar

The Program of the Antarctic Syowa MST/IS radar (PANSY) is a Mesosphere–Stratosphere–Troposphere/Incoherent Scatter (MST/IS) radar located at the Japanese Syowa Station (69.01°S, 39.59°E) in the Antarctic (Sato et al., 2014). The first subarrays
of the PANSY radar were installed in 2011. The first continuous observations of Polar mesospheric summer echoes were made with 19 subarrays in January-February 2012. Due to snow accumulation in the originally symmetric antenna field consisting of 1045 crossed Yagi antennas summed into 55 channels, several of the subarrays were moved to higher ground as illustrated in Fig. 3. This is the antenna configuration we have used in the simulations.

PANSY operates on a center frequency of 47 MHz and with a peak power of 500kW and 5% duty cycle. The radar is a
challenge for DOA determinations as the subgroups are located at different altitudes and partially disjoint, and have to be moved or intermittently be disconnected from the system depending on snow accumulation conditions. Even the antennas within subgroups are elevated non-symmetrically. Currently the antennas are distributed in altitudes ranging between -2 and +8 meters from the reference plane.



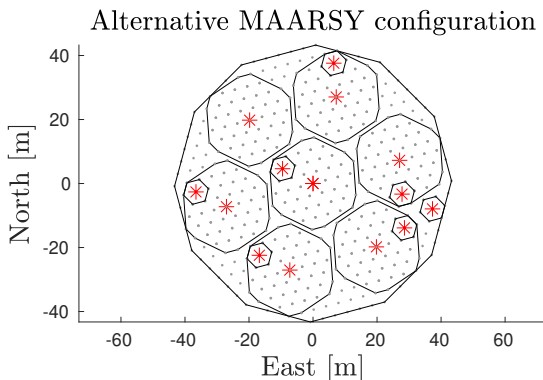

**Figure 4.** An alternative configuration of MAARSY subgroups used as radar channels to the one illustrated in Fig. 3. In this configuration 15 channels are used instead of 8 as to include the information from smaller but closely located hexagonal groups. Thus producing shorter baselines for less ambiguous interferometry (Schult et al., 2017).

In 2017, a peripheral antenna array for detecting field-aligned irregularities (FAI) were installed (Hashimoto et al., 2019). This has enabled suppression of FAI echoes and increased the number of power profiles usable for incoherents scatter measurements of the polar ionosphere by more than 20%. In this paper we do not investigare the peripheral FAI array.

### 4.5 PANSY meteor radar

The PANSY radar has recently been complemented by a meteor trail echo interferometric receiver system (Taishi Hashimoto, personal communications). The antenna configuration is displayed in Fig. 3. Since the operating frequency of PANSY (47 MHz) differs from meteor radar systems (typically 35 MHz), the configuration is more compact when displayed in units of metres even though the number of wavelengths are the same. The main difference between the PANSY meteor radar receiver and the Jones $2.5\lambda$ radar is instead that the prior is not a planar array but that the antennas are displaced in the vertical (z) direction with up to 0.8 m, corresponding to $\sim 0.12\lambda$.

## 5 Results

To demonstrate the above methods we present results from numerical simulations. The next step will be applying them on measurement data. We aim to implement these methods in our data analysis pipelines for meteor head echos measured by the MU radar and the PANSY radar in the future, as well as classify the location probability of ambiguous meteor radar trail echoes using Bayesian inference. However, the current study allows us to quantitatively evaluate how DOA determination behave with respect to SNR and qualitatively evaluate if ambiguities are relevant or not. Such results are useful in configuration and construction of pipelines.





For each of the radar systems described in Sect. 4 we have applied the methods described in Sect. 2 and 3. Three input directions $\mathbf{k}_0$ were chosen as sources:

   I. $\mathbf{k}_I$ = Azimuth: $0°$, Elevation $75.5°$,

   II. $\mathbf{k}_{II}$ = Azimuth: $0°$, Elevation $90°$,

III. $\mathbf{k}_{III}$ = Azimuth: $45°$, Elevation $40°$.

For each of these chosen sources the following steps were performed:

– Determine all ambiguities using 1000 starting conditions according to the method outlined in Sect. 2.1. This generates the $\Omega_X$ and $\Omega_Y$ sets.

– Run an MC simulation of 500 samples for each input direction in $\Omega_X$ at all SNR levels, according to the method outlined
in Sect. 2.4. An appropriate range of linearly spaced SNRs in decibel was used to capture the transition from stable DOA determination to complete algorithm failure.

– Discretise the MC results into probability matrices $P_{ij}$ using the sets $\Omega_X$ and $\Omega_Y$ according to Sect. 2.5, using an inclusion radius of $s = 0.07$.

– If applicable, simulate measurements according to Sect. 2.7 and calculate Bayesian inference distributions according to
Sect. 2.6.

## 5.1 Jones 2.5λ radar

We begin with the Jones 2.5 λ radar as this is the simplest system examined in this study. We provide more detailed results and examples for this system to illustrate the methodology. For the remaining systems we only provide summary results.

Following the list outlined previously, for $\mathbf{k}_I, \mathbf{k}_{II}$ and $\mathbf{k}_{III}$ the resulting DOA sets $\Omega_X$ and $\Omega_Y$ are illustrated in Fig. 5. Here
the top row of color-maps indicates the normalized sensor response distance as defined in Eq. 6, where the red dots indicate $\mathbf{k}_0$. The cyan crosses indicate the simulated input DOAs as described by $\Omega_X$. The large red circles indicate the inclusion regions with radius $s = 0.07$ for discretisation determined by the output set $\Omega_Y$. The bottom row consists of maps of input-output locations where input indices refer to elements in $\Omega_X$ and output indices refer to elements in $\Omega_Y$. These indices are used in subsequent results.

As expected, Fig. 5 indicate that the Jones configurations has prominent ambiguities at $\pm 0.43$ in the directional cosine along either of the array axis. As this is a simple radar system, these ambiguities can be found with conventional methods equivalent to the Nyquist-Shannon sampling theorem (Jones et al., 1998).

Using the $\Omega_X$ set, a series of MC simulations were performed. One of these simulations at $SNR = 7.24$ dB for $\mathbf{k}_I$ is illustrated in Fig. 6. discretising all the MC simulations produced a series of $P_{ij}$ matrices. The one corresponding to Fig. 6 is
given as an explicit example in Table. 1.





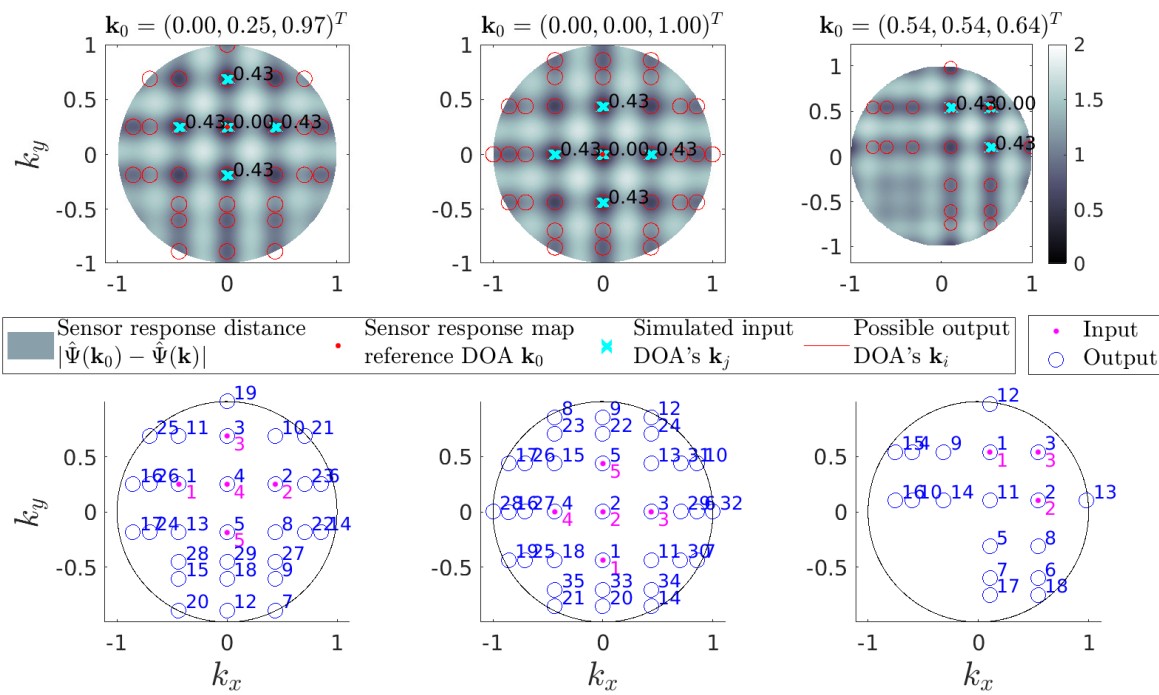

**Figure 5.** Ambiguity analysis summary illustration for the Jones 2.5 $\lambda$ radar. The three columns, from left to right, represent the source DOAs $\mathbf{k}_I$, $\mathbf{k}_{II}$ and $\mathbf{k}_{III}$ presented at the beginning of Sect. 5. For each of the three different source DOAs $\mathbf{k}_0$, the top row consists of sensor response distance $d(\mathbf{k})$ calculated using Eq. 6. Overlaid on the map are markings according to the legend: the reference DOA $\mathbf{k}_0$, the simulated input DOA set $\mathbf{k}_j \in \Omega_X$ and the possible output DOA set $\mathbf{k}_i \in \Omega_Y$. The sets are calculated as described in Sect. 2.5. The bottom row shows an indexing map for Input-Output locations that will later be used to illustrate the data.

Tables such as Table. 1 are not efficient at conveying the probabilistic information. Instead we illustrate the matrices $P_{ij}$ as a function of SNR in Fig. 7 for the source $\mathbf{k}_I$. In this figure one panel illustrates a column of the $P_{ij}$ matrix as a function of SNR where each curve represents a row of the matrix, as described by the figure legend. The results for $\mathbf{k}_{II}$ and $\mathbf{k}_{III}$ are practically identical to the ones illustrated in Fig. 7, but shifted in SNR space. Their relative shift can be seen in the summary of all results in Table 2.


These illustrations show the region where noise-induced ambiguities are relevant. For example, for Input 1 in Fig. 7 the DOA is always correctly determined above 10 dB SNR. Noise-induced ambiguous solutions appear between -10 dB and 10 dB SNR. At lower SNR, the algorithm returns approximately uniformly distributed results classified as algorithm failure. However, for Input 3 the onset of noise-induced ambiguities occurs below 18 SNR dB and total failure below -3 SNR dB. This is partially

due to the full array gain being 4.3 dB larger at Input 3, thus the individual channel SNR is lower for this DOA.





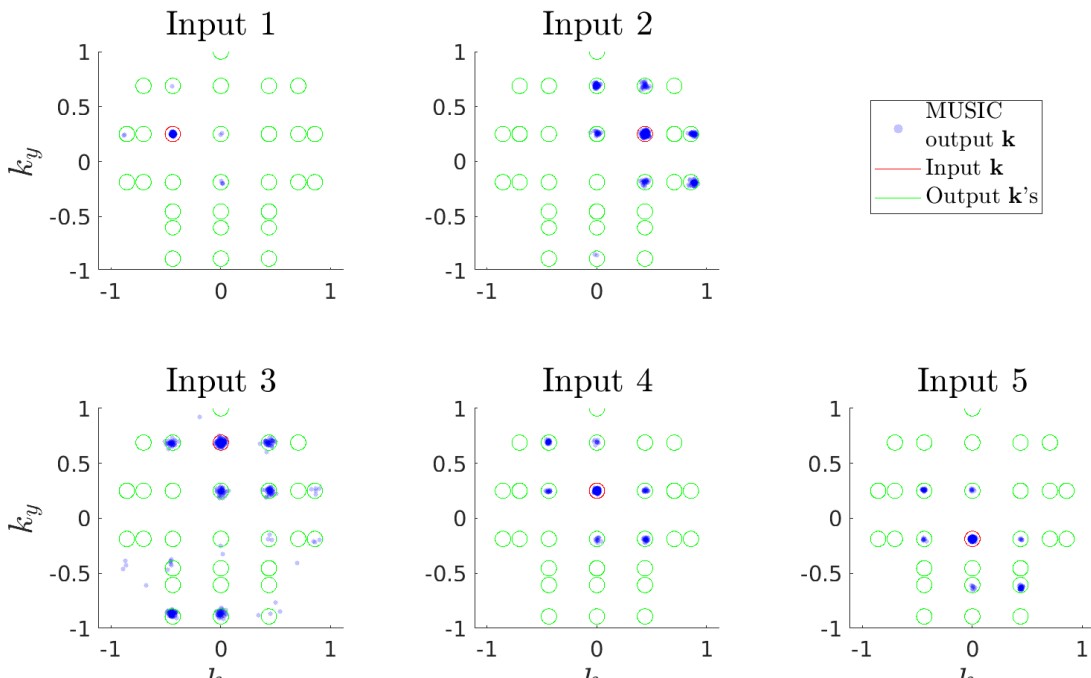

**Figure 6.** Five MC simulations with 500 samples each using an inclusion radius of $s = 0.07$ for the Jones $2.5\lambda$ radar. Each panel illustrates a simulation using an element of $\Omega_X$ (marked with a red circle in each panel) as input. In all cases the SNR was set to 7.24 dB SNR after coherent integration over all channels. The DOA determination dynamics are different for each input location in this example as the DOA determination depend on the channel SNR while the coherently integrated SNR was fixed – the ratio between the coherently integrated SNR and the channel SNR depend on DOA. For each location, 500 simulated noisy data sets were produced and analysed using MUSIC DOA determination. These DOA outputs are marked as transparent blue dots and the expected possible output directions $\Omega_Y$ are marked in each panel by green circles.

As meteor trail echoes are generally not observed at the zenith, an array SNR of 10 dB is sufficient to determine the DOA correctly for the majority of the observed events, as was reported also by Jones et al. (1998). However, different input directions have different thresholds. For most directions 10 dB is not sufficient for 99% confidence. The information provided by the simulations may be used to reliably determine the true input DOA of events much lower than 10 dB.

Following the method outlined in Sect. 2.6, we can generate a simulated series of observations and analyse that series with Bayesian inference to find $\mathcal{P}_x(\mathbf{k}_j)$. Such a simulation and analysis using source $\mathbf{k}_I$ is illustrated in Fig. 8. Here, ten observations were generated by the probabilities from Input 4, i.e. column 4 of the $P_{ij}$ matrix, at 5.52 dB SNR. Outputs labeled $F$ indicate "algorithm failure".

The interesting aspect of this simulated example is how the Bayesian approach incorporates the information from the simu-
lations to make inferences. For example, already after three observed outputs we are able to quantitatively identify Input 4 as



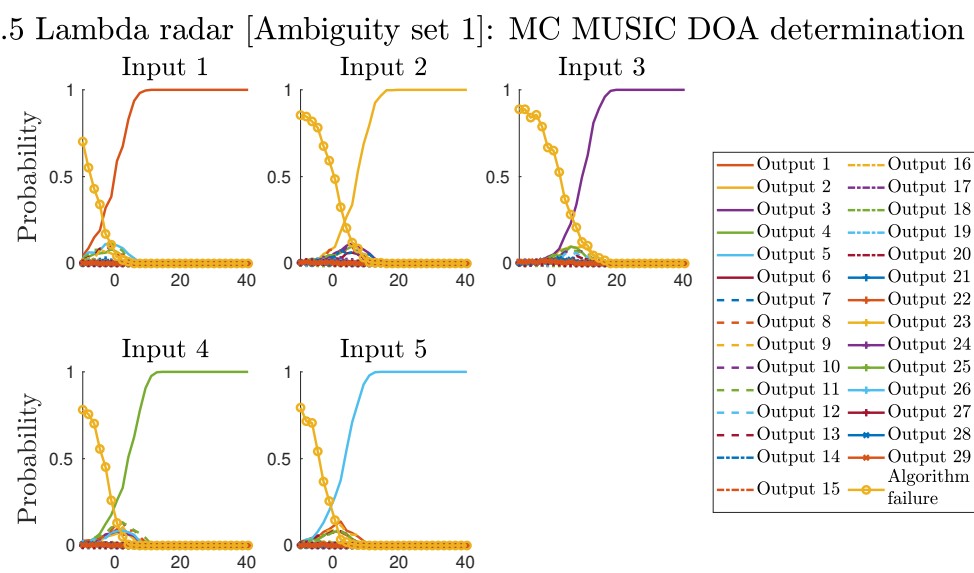

**Figure 7.** The discretised output DOA distribution as a function of SNR and input DOA for the Jones $2.5\lambda$ radar. The simulations were made with respect to the ambiguity sets calculated for $\mathbf{k}_I$. Each panel corresponds to a certain simulated input DOA. The curves in each panel correspond to the probability of the MUSIC DOA determination producing that DOA as an output. The indexing of the Inputs and Outputs follows the bottom left Input-Output indexing map of Fig. 5. If discretised, the MC simulation illustrated in Fig. 6 correspond to a vertical slice at around 7 dB SNR. Thus, the same slice also correspond to Table. 1. The DOA outputs that do not fall into any discretisation region are considered algorithm failure.

the most probable source with high certainty, even though it was directly observed as an output in just half of the ten samples. We cannot expect such perfect inference when applied to real data as models are not perfect at describing data, but this approach should anyway provide a reasonable quantitative confidence of the true DOA. Once a probable true input location has been identified, the DOA determination can be re-done but restricted to solutions in a non-ambiguous region, thus allowing the

originally "bad" output to be used.

Before application on measurement data one should validate that the $\Omega_X$ and $\Omega_Y$ sets are predicting the behaviour of the MC simulations so that unexpected dynamics introduced by the DOA determination algorithm itself are not disregarded. If the measurement data cannot be explained by MC simulation, the sensor response model or the phenomenon model are most likely not representative.

Considering these examples, we examine how often the Bayesian inference is able to correctly identify the true input. This is done by repeating the simulation illustrated in Fig. 8 many times and calculating the probability that the true input is assigned the highest probability by Eq. 21. Repeating this process for every SNR level that was simulated we find the probability of correct classification as a function of SNR and number of observed outputs. This relation is illustrated in Fig. 9 for the inputs $\mathbf{k}_I$, $\mathbf{k}_{II}$ and $\mathbf{k}_{III}$, respectively. Here the DOAs $\mathbf{k}_I$ = Input 4, $\mathbf{k}_{II}$ = Input 2 and $\mathbf{k}_{III}$ = Input 3 were chosen as the true



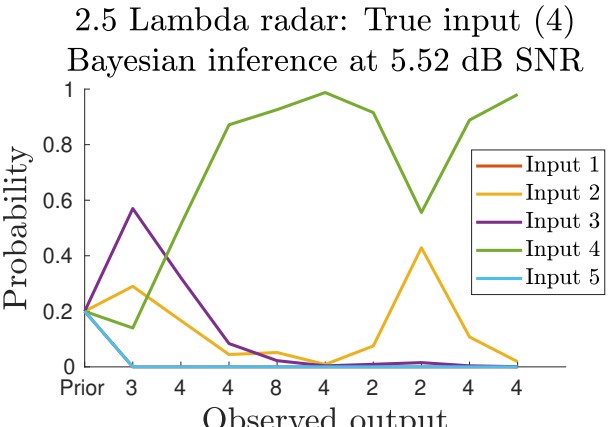

**Figure 8.** A simulated series of observed output locations based on the discretised MC DOA determination. A true input and an SNR level is chosen, in this case Input 4 from the $\mathbf{k}_{II}$ simulations at 5.52 dB SNR. The appropriate probability matrix $P_{ij}$ was then chosen (similar to the example given in Table. 1) to generate random observations. The probability of observing Output N is given by row N and column 4, the observations are thus drawn from a multinomial distribution equal to column 4. The indices follow the bottom left Input-Output map illustrated in Fig. 5. $\Omega_Y$ consists of 29 elements and "Algorithm failure" is labeled as output $F$. The output is analysed using Bayesian inference according to Eq. 17 to find the probability of which input is the true one.

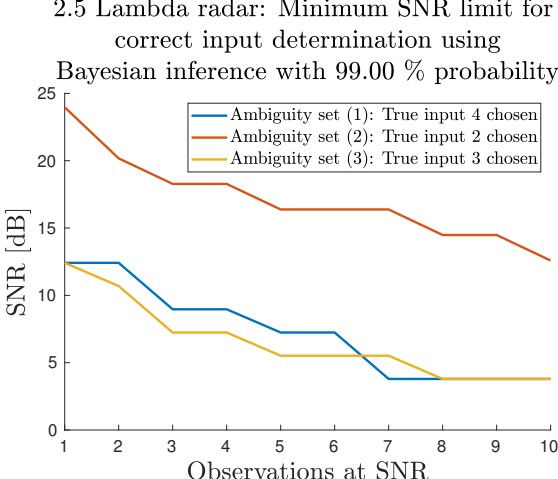

**Figure 9.** Minimum SNR needed to correctly identify the true input direction in 99% of all cases using Bayesian inference, as a function of number of observations. This is of course assuming the models represent reality. These curves are calculated by repeatedly simulating observations such as the ones illustrated in Fig. 8. Using this set of simulated observations and subsequent analysis gives a probability of correct identification. For example, in Fig. 8 the correct input is identified with >80% probability after the seventh observed output. This procedure is performed for all SNR levels and for the three cases $\mathbf{k}_I, \mathbf{k}_{II}$ and $\mathbf{k}_{III}$ to compile the three curves illustrated here.





DOA for the measurement simulation. The illustration shows the minimum SNR needed for the inference to assign the largest probability to the true DOA in 99% of all cases, as a function of observed outputs. This relationship indicates which type of events that can be confidently analysed for this particular radar system. For example, consider a Jones 2.5 $\lambda$ radar that observed seven independent echoes at 6 dB SNR from the same meteor trail. In this case, Fig. 9 indicates that the direction given the largest probability by Eq. 21 is a reliable choice.

If one also considers population distributions, it is known that meteors occur mostly between 70–130 km altitude (Kero et al., 2019, and references therein). Let us consider the ambiguities illustrated in the left column of Fig. 5 and assume the the target is at a range of 98 km. Then, the $\mathbf{k}_0$ that is located at 75.5° elevation, marked with a sensor response distance $d = 0.0$ and a red dot, is at a height of 95 km. This range produces 96 km height at ambiguities of distance $d = 0.96$, and 86 km or 57 km height at distances $d = 1.36$. Thus, in such a scenario one could not rule out these ambiguities due to altitude. However, if

instead $\mathbf{k}_0$ would be located at 70 km altitude, any other location would be improbable, yet still not impossible. Fireballs have been detected by standard meteor radar systems (Schult et al., 2015). If such an event would appear with suitable geometrical alignment with respect to the radar, it could be detected in a far side-lobe as low down in the atmosphere as 40 km where ablation typically ends.

In the case of a $\mathbf{k}_0$ in zenith, as illustrated in the middle column of Fig. 5, this is an unlikely event with respect to specular

trail echos as this would require the meteor to travel perpendicular to the ground. In this case, if we assume that the target is at a range and height of 95 km, then the ambiguities labeled $d = 0.96$ are located at 85 km height and are thus also reasonable with respect to the population altitude distribution.

Lastly, we consider the ambiguities illustrated in the right column of Fig. 5 where $\mathbf{k}_0$ = Azimuth: 45°, Elevation 40°. If an object is located at the $\mathbf{k}_0, d = 0.0$ direction and we assume a range of 132 km, the target is located at an altitude of 85 km.

The ambiguities labeled $d = 0.96$ and $d = 1.36$ are then located at a height of 110 km and 130 km.

There are many arguments that can be made, similarly to the ones described above, to restrict the possible directions to regions that do not include any ambiguities. But, while doing such considerations one should also take range aliasing into account. If the commonly used pulse repetition frequency 2144 Hz of the SKiYMET radar systems is used, a range aliasing of $\simeq 70$ km is present (Hocking et al., 2001). Such problems can of course easily be solved by coded transmission sequences

thereby removing range aliasing (Vierinen et al., 2016).

## 5.2 MU radar

In contrast to the Jones $2.5\lambda$ radar, the MU radar channels consists of subgroups of antennas. If all subgroups were identical and had a reflection symmetry-line (this mathematically assures that the subarray gain does not have an imaginary component) it would be the same situation as with the antenna gain function $\gamma$ which means that the subarray gain patterns $g_j$ could be

omitted. However, as is illustrated in Fig. 3, the MU radar has six outer subgroups that are not symmetric. Thus, the subgroup gain will affect the normalized sensor response model and the DOA determination capabilities. Therefore we consider both the models described in Eqs. 29 and 30.



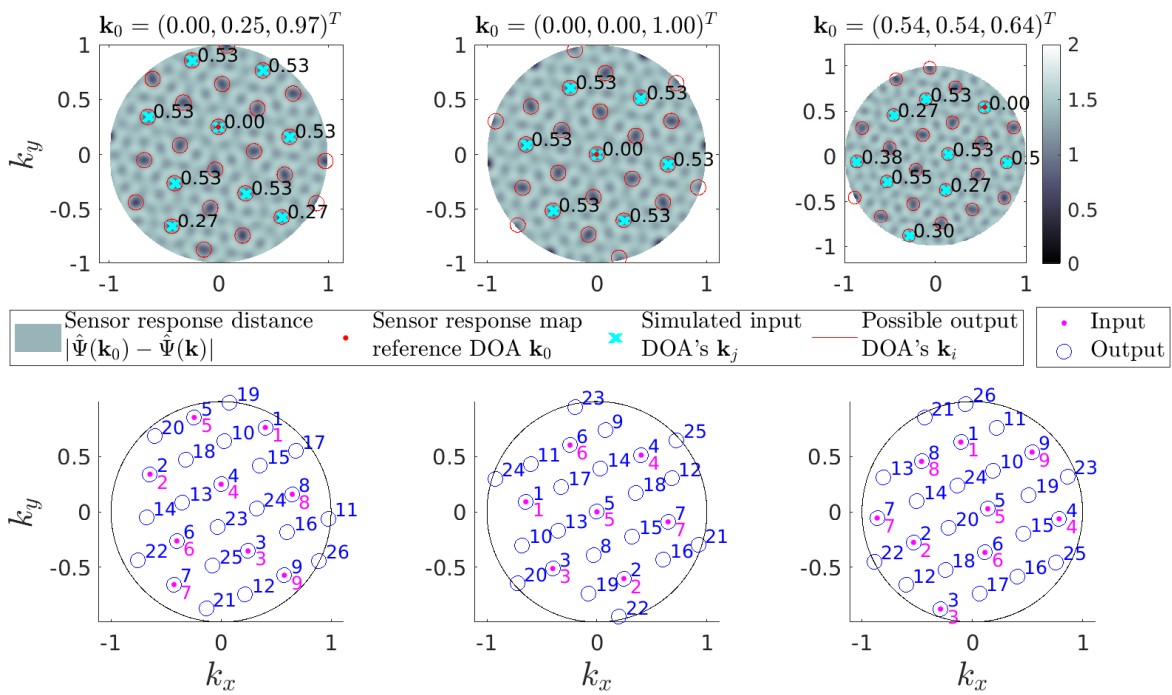

**Figure 10.** Ambiguity analysis summary illustration for the MU radar using the phase center model. A detailed explanatory caption is given in Fig. 5.

Given the MU asymmetric subgroups, one could consider the model in Eq. 29 unphysical. Nevertheless, the phase center model has been successfully used to analyse meteor head echoes from the MU radar (Kero et al., 2012b). The reason is that the

two models obviously converge towards the zenith and are very similar in the main lobe and first side-lobes as they are both models of planar arrays. Most meteor head echo detections occur in the main lobe of the radar so only a small portion of the events are affected by the difference between the two models.

     The ambiguities formed for $\mathbf{k}_I$, $\mathbf{k}_{II}$ and $\mathbf{k}_{III}$ using the phase center model and the resulting DOA sets $\Omega_X$ and $\Omega_Y$ are illustrated in the the top row of Fig. 10. Their Input-Output index relations are illustrated in the bottom row. The respective

results using the subgroup model are illustrated in Fig. 11.

     A few important observations regarding the ambiguities for the phase center model: there are no ambiguities with $d = 0$, they are symmetrically spaced and co-moving with the source point. This is expected when considering that 19 of the subgroups are positioned in a hexagonally symmetric pattern and thereby should form perfect ambiguities. These subgroups are the ones that create the symmetric pattern. The six asymmetric groups are breaking the symmetry sufficiently to distinguish between these

perfect ambiguities, thereby creating the patterns seen in the top row of Fig. 10. This indicates that the noise in the respective symmetry-breaking subgroups determine if the DOA can be determined.





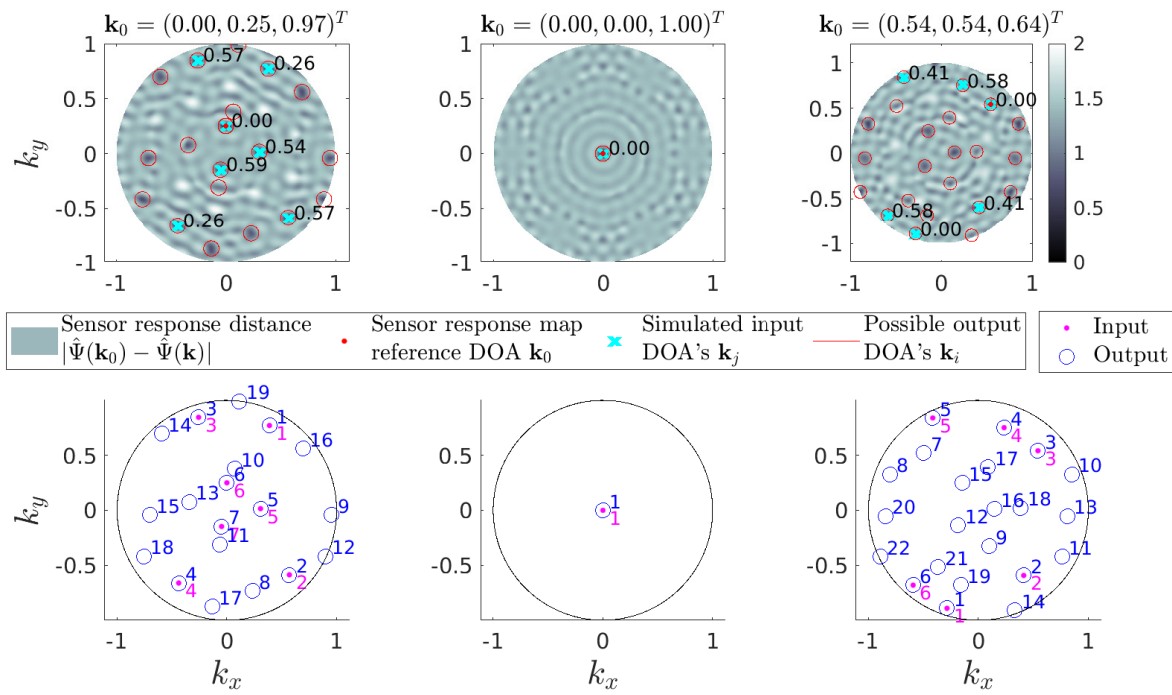

**Figure 11.** Ambiguity analysis summary illustration for the MU radar using the subgroup model. A detailed explanatory caption is given in Fig. 5.

Comparing the phase center model with the subgroup model, the DOA determination situation improves slightly for $\mathbf{k}_I$ and $\mathbf{k}_{II}$ as many ambiguities become less prevalent. This can be attributed to the fact that the two models diverge at lower elevations, thereby making it easier to distinguish a low-elevation DOA from a DOA near the zenith. However, the closest

ambiguities basically remain at the same distance $d = 0.26$. Furthermore, for $\mathbf{k}_{III}$, there is even a close to perfect ambiguity ($d = 0.00474$) as illustrated by the top right panel in Fig. 11. This may appear surprising but makes sense given the internal antenna configuration of the outer subgroups that differentiate between the ambiguities formed by the 19 inner subgroups.

The phase output of subgroup 1, the outer subgroup to the west in Fig. 3, using the subgroup model versus the phase center model is illustrated in Fig. 12. The two red crosses correspond to the two directions labeled $d = 0$ in the top right panel in

Fig. 11. Here it is clearly seen that the phase have equal values for these two directions when including subgroup gain in the left panel, while the phase values were very different when using the phase center approximation shown to the right. This channel is the main contributor to differentiating between these two ambiguities that appear due to the 19 inner subgroups. This highlights the advantage of numerically investigating ambiguities, either by sensor response model distance maps or by MC simulation as this ambiguous DOA would probably have gone unnoticed otherwise.



**Figure 12.** Comparison of the signal phase measured between the subgroup model and the phase center model. Illustrated is the phase measured for the MU radar channel 1, i.e. the outer asymmetric subgroup to the west illustrated in Fig. 3. The inclusion of the asymmetric antenna positions in the subgroup model affects the expected phase measurements of the signal as a function of wave DOA. The two red crosses mark two direction that are ambiguous in the subgroup model but not in the phase center model.

The above results raise the question of which directions can be determined by the MU radar using the more realistic subgroup model. To answer this question we performed a sweep of the ambiguity searching algorithm outlined in Sect. 2. Instead of only using $\mathbf{k}_I$, $\mathbf{k}_{II}$ and $\mathbf{k}_{III}$, a grid of $\mathbf{k}$-vectors over all possible directions was used. On each grid-point all ambiguities were calculated and the minimum distance was saved for that direction. For example, for $\mathbf{k}_I$ the distance $d = 0.26$ from the top left axis in Fig. 11 would be saved and for $\mathbf{k}_{III}$ we would save $d = 0.00474$, according to the top right axis in Fig. 11. The resulting

map is illustrated in Fig. 13. It shows which source directions the MU radar is able to resolve well and which directions it cannot uniquely determine. The white areas are regions where no ambiguities were found using the selected algorithm settings. Since a minimum distance of 0.6 was set for saving ambiguities, none are found if they all have distances larger then this number. For convenience, two elevation limits are also drawn onto the minimum distance map. These show that any direction above 48° elevation can be determined, with varying robustness, and that any direction above 81° elevation is always very well

determined.

Figure 14 contains an MC simulation using the phase center model at an SNR value of 2.76 dB. This example illustrates the good correlation between the $\Omega_X$ and $\Omega_Y$ sets generated by the ambiguity analysis and the MC simulations.



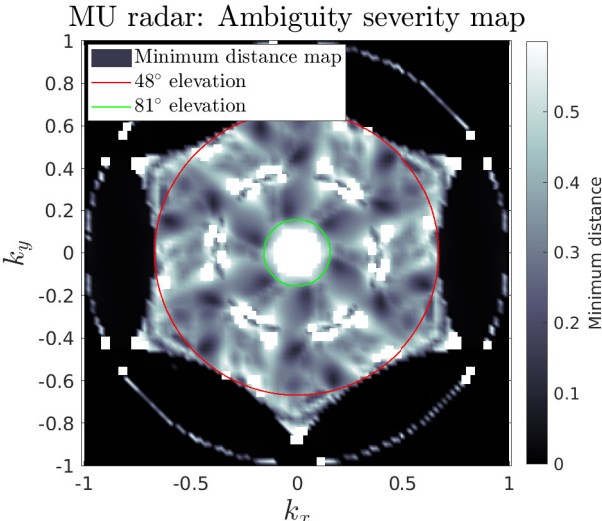

**Figure 13.** Summary results from running the ambiguity analysis outlined in Sect. 2.1. The ambiguity analysis ran using only first order ambiguities for the dense grid of possible DOAs illustrated here. The mapping shows the minimum distance $d$ from the resulting ambiguous directions at that input DOA location. For reference, two elevation limits at $48°$ and $81°$ are added to the illustration. These indicate above which elevations we expect DOA determination with the MU radar to be unambiguous and robust, respectively.

In Fig. 15 the summary of the MC MUSIC DOA determinations is illustrated for $\mathbf{k}_{II}$. The Input and Output labels in these illustrations correspond to the map of locations in the bottom row of Fig. 10. The dynamics of the DOA determination shown here is as expected when using the simplified model of the sensor response. The dynamics for $\mathbf{k}_I$ and $\mathbf{k}_{III}$ are practically identical but shifted in SNR space. The respective shift of each MC simulation can be seen in Table 2. It should be noted that the SNR necessary to correctly determine the DOA differs between $\mathbf{k}_I$ and $\mathbf{k}_{II}$ up to 20 dB for many of the inputs. This suggests that one should not adopt a single full array SNR threshold for discarding events but rather: a channel SNR threshold; a DOA output dependant threshold; a MUSIC peak value threshold; a combination of the above.

The phase center model results should only be viewed as informative simulations, not as a predictors for real behaviour. To infer information about how real measurements behave we should focus on the MC simulations of the subarray model. In Figs. 16, 17 and 18 the summary of the MC MUSIC DOA determinations for that model is illustrated for $\mathbf{k}_I$, $\mathbf{k}_{II}$ and $\mathbf{k}_{III}$, respectively.

Examining Fig. 16 and cross-referencing the input locations with the Input-Output map in Fig. 11 we note several interesting results.

Firstly, we see trouble determining directions uniquely if they are in the "dark-zone" indicated by Fig. 13, i.e. panels labeled Input 1 trough 4. Here the MUSIC DOA determination is unable to select the correct output for Input 3 and 4 even though there is a small difference between the signals ($d = 0.00474$ as mentioned previously). As these simulations were using a single



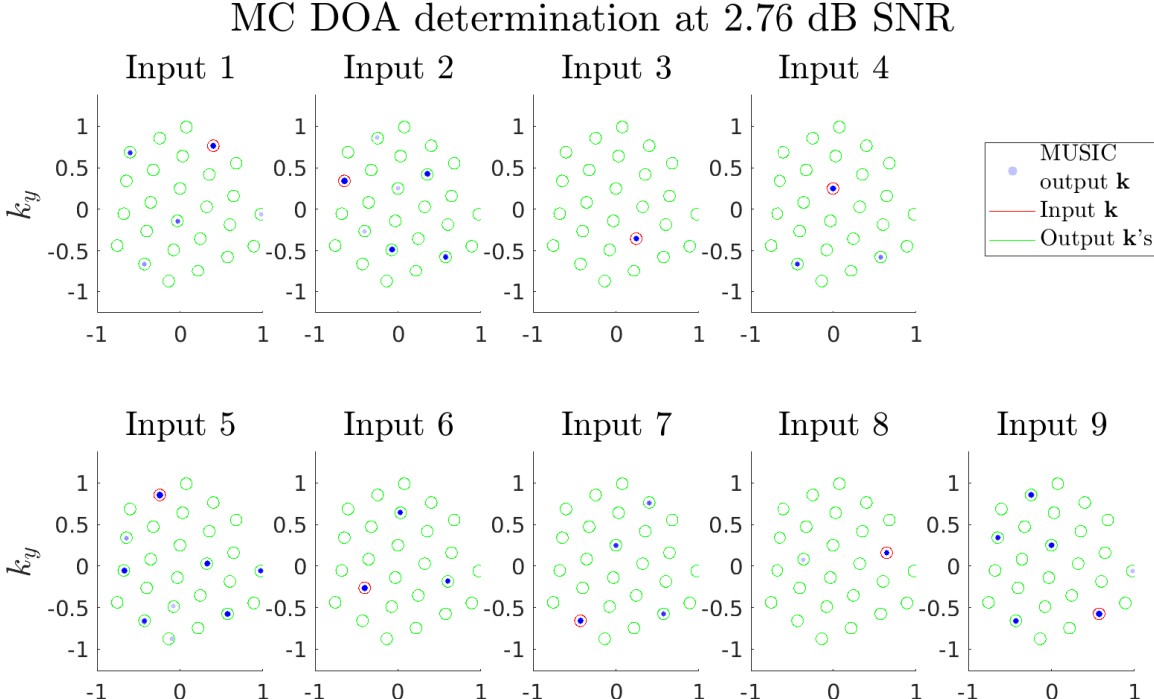

**Figure 14.** Nine MC simulations with 500 samples each using an inclusion radius of $s = 0.07$ for the phase center MU radar model. A detailed explanatory caption is given in Fig. 6.

starting point for the MUSIC gradient ascent, the behaviour could be caused by a narrow-peak problem. When using a fixed
grid to select the start point for the MUSIC gradient ascent it might miss a narrow peak.

Secondly, for sources inside the unambiguous region, i.e. Input 5 through 7, the DOA determination works as expected, with an algorithm failure onset slightly below 0 dB SNR. But compared to the results for the phase centre model, ambiguities now seem irrelevant as there is hardly any ambiguous DOA that is assigned a high probability.

As indicated by Fig. 13, all directions located close to the zenith are very robustly determinable. Therefore, no extensive MC
simulations are needed for $\mathbf{k}_{II}$. Instead, we ran a set of sparse simulations in the SNR space to examine the onset of algorithm failure, illustrated in Fig. 17. In this case the onset occurs below around 12 dB SNR.

The different input locations differ significantly in the SNR needed for stable DOA determination. This limit also differ with respect to the used sensor response model. As such, the MUSIC peak value is a more stable quality indicator than SNR for DOA determination. The MUSIC peak value directly describes how well the used sensor response model matches the
measured signal. The MUSIC peak distribution for the MU subarray model simulations is illustrated in Fig. 19. These are the same simulations that are illustrated in Fig. 17. This distribution is useful for validating sensor response models. If measured

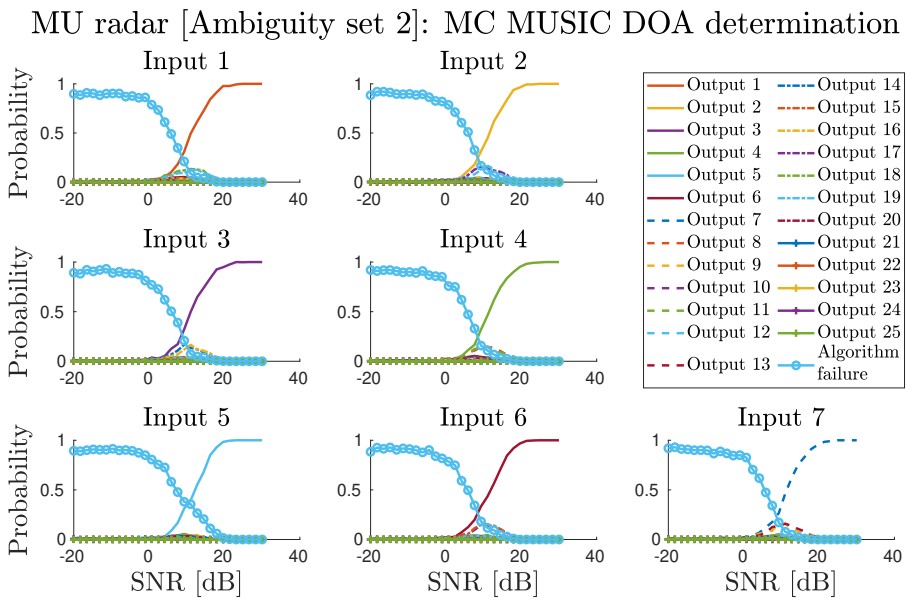

**Figure 15.** The discretised output DOA distribution as a function of SNR and input DOA for the MU radar. The simulations were made with respect to the ambiguity sets calculated for $\mathbf{k}_{II}$ using the phase center model. A detailed explanatory caption is given in Fig. 7.

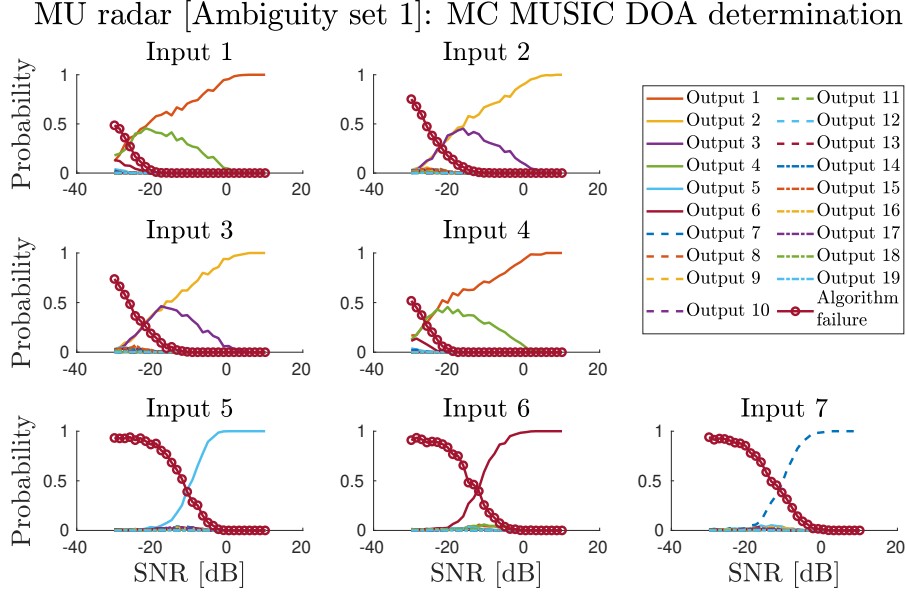

**Figure 16.** The discretised output DOA distribution as a function of SNR and input DOA for the MU radar. The simulations were made with respect to the ambiguity sets calculated for $\mathbf{k}_I$ using the subgroup model. A detailed explanatory caption is given in Fig. 7.





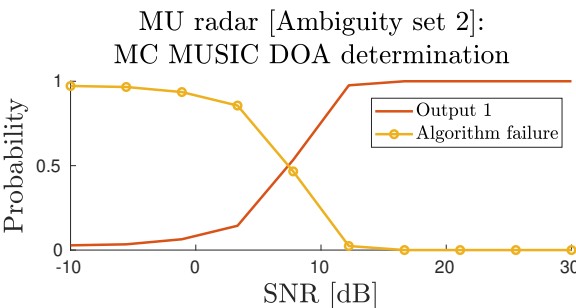

**Figure 17.** The discretised output DOA distribution as a function of SNR and input DOA for the MU radar. The simulations were made with respect to the ambiguity sets calculated for $\mathbf{k}_{II}$ using the subgroup model. A detailed explanatory caption is given in Fig. 7.

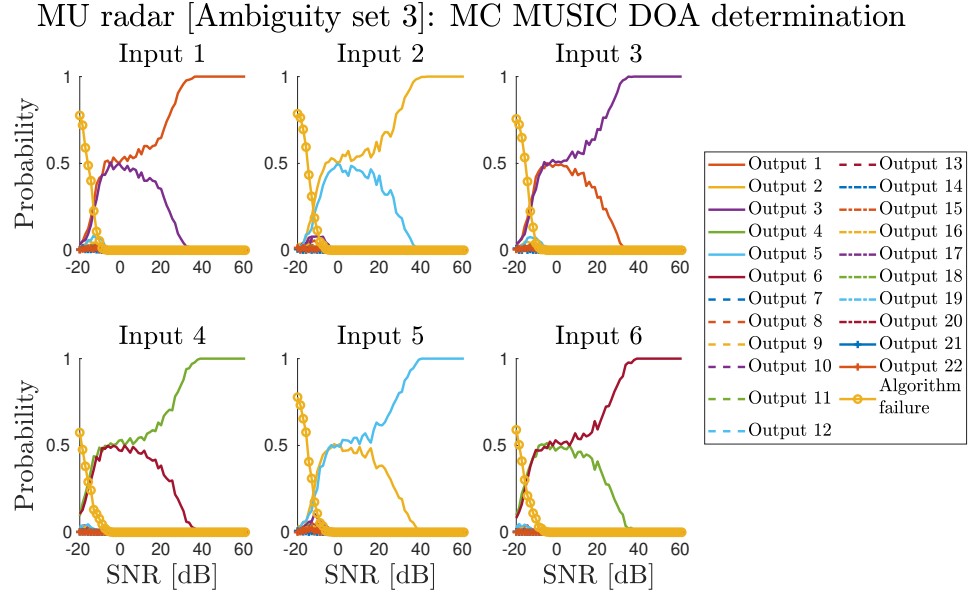

**Figure 18.** The discretised output DOA distribution as a function of SNR and input DOA for the MU radar. The simulations were made with respect to the ambiguity sets calculated for $\mathbf{k}_{II}$ using the subgroup model. A detailed explanatory caption is given in Fig. 7.

echoes deviate from this distribution, either the DOA was not correctly determined or it indicates an erroneous sensor response model.

As mentioned and shown by Inputs 1 through 4 in Fig. 16, there are problems with correctly determining DOA in the 600 ambiguous region even though it should in principle be possible at a high enough SNR. To test if launching several parallel gradient ascents as described in Sect. 3 would improve algorithm robustness, we have performed additional simulations for $\mathbf{k}_{III}$.





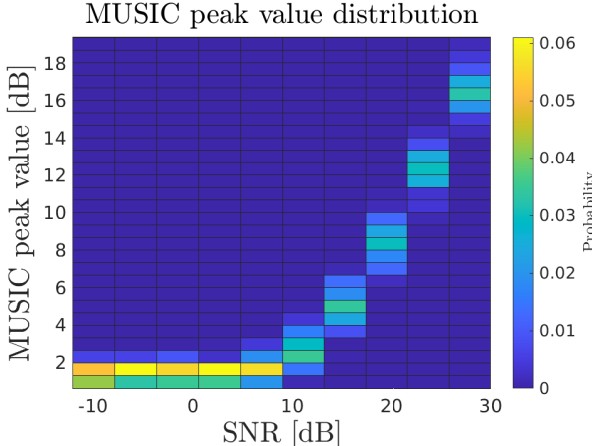

**Figure 19.** Distribution of MUSIC peak values as a function of SNR compiled from all MC simulation with the zenith as the input DOA, i.e. from the simulations illustrated in Fig. 17. This example is for the MU radar using the subarray model described in Eq. 30. When the MUSIC peak value flat-lines the algorithm can no longer find any significant matches and the output DOA is approximately uniformly distributed. The region just before the flat section of the distribution is where ambiguities can occur. As a reference, the stable DOA determination occurs for SNRs above 12 dB for this input DOA.

In the additional simulations we chose to take the $N = 20$ largest grid points with a minimum separation of $\delta X = 0.1$ as initial conditions for the gradient ascents. We also increased the examined SNR range to find the region within which a
$d = 0.00474$ ambiguity could be correctly determined. The resulting MC statistics are illustrated in Fig. 18.

The DOA determination dynamics can be explained as follows: around and below -10 dB SNR the complex noise perturbs the signal through the sensor response model surface and the distribution becomes uniform as a result of projection, i.e. algorithm failure. Between around 0 and 20 dB SNR, the mean noise perturbation distance decreases and it penetrates less of the sensor response model surface, except for two close output points. Thus, they follow in tandem until we reach a break-point, at around
30 dB SNR. Above this SNR level the perturbation in sensor response space decreases below the $d = 0.00474$ limit. Finally, it stabilises above around 40 dB SNR when the perturbation is too small to reach any other surfaces of the sensor response space. This example shows that for targets with SNR higher than 40 dB, an unambiguous DOA determination in this region is possible. Candidates for producing such strong echoes include bolides with large radar cross section and active satellite.

As the MU radar is a more complex system then the Jones $2.5\lambda$ radar it might be misleading to simulate Bayesian inference.
Also, the results indicate that for many directions ambiguities are not relevant. More information about how well the model in Eq. 30 describes the behaviour of the radar response at all elevations is required. As a next step we intend to closely examine real data to determine if the models are realistic enough to allow Bayesian inference of ambiguous events.

Since the sensor response models described in Eqs. 29 and 30 converge at zenith, the MC simulation results becomes an indication of which sensor response model is the better choice. In Figs. 10 and 11, the subgroup model performs better as it
differentiates between directions much better then the phase center model. In Fig. 15, the Input 5 is located at the zenith and





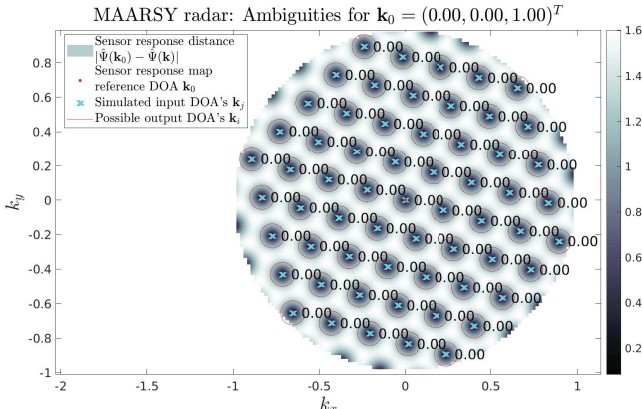

**Figure 20.** Ambiguity analysis summary illustration for the MAARSY-8ch phase center model. The ambiguity analysis represent the source DOA $\mathbf{k}_{II}$, i.e a DOA in zenith. The illustration consists of a sensor response distance $d(\mathbf{k})$ map calculated using Eq. 6. Overlaid on the map are markings according to the legend: the reference DOA $\mathbf{k}_0$, the simulated input DOA set $\mathbf{k}_j \in \Omega_X$ and the possible output DOA set $\mathbf{k}_i \in \Omega_Y$. The sets are calculated as described in Sect. 2.5.

displays an algorithm failure onset below around 20 dB SNR. However, in Fig. 17 this onset only occurs below around 12 dB SNR. This indicates a significant improvement in the SNR limit for analysing events if the subgroup model is used.

## 5.3 MAARSY radar

The MAARSY radar system is limited to 16 output channels but with flexible subgroup configuration. Studies looking at
interferometry of meteor echoes with MAARSY have predominantly used two different configurations. These configurations are illustrated in Figs. 3 and 4 and we hereafter refer to the them as **MAARSY-8ch** (Schult et al., 2013) and **MAARSY-15ch** (Schult et al., 2017).

The phase center model of the MAARSY-8ch configuration contain perfect ambiguities as illustrated in Fig. 20. In this antenna configuration all subgroups contain reflection-symmetry lines, which means that their individual subgroup gains do
not resolve ambiguities. However, the MAARSY-8ch configuration also contains the entire array as one of the channels. This channel has a significantly different gain pattern compared to the other channels. This creates a small shift in the sensor response between different directions. We have not included an illustration of the ambiguity analysis of the subgroup model for the MAARSY-8ch configuration. Its ambiguity map is similar to the one illustrated in Fig. 20 with the main difference that most of the ambiguities have a distance between 0.07–0.25 while a few remain close at $d = 0.01$ instead of being perfect.
For the phase center model of the MAARSY-15ch configuration there are many close to perfect ambiguities and a few perfect ones. The distribution of ambiguities is close to identical to the one illustrated in Fig. 20. The addition of the smaller hexagonal subgroups, illustrated in Fig. 4, creates a decent basis for being able to determine a trajectory uniquely. In practice, the results indicate that this would work reliably for very high SNR targets only.





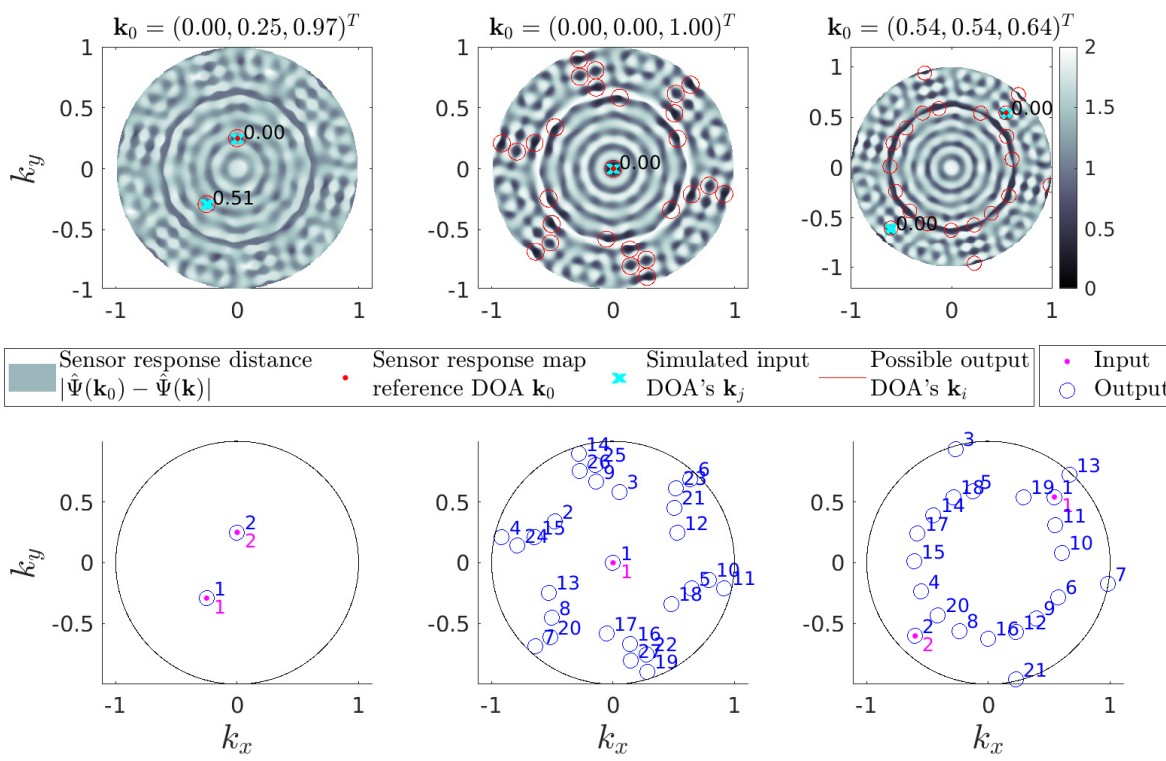

**Figure 21.** Ambiguity analysis summary illustration for the MAARSY-15ch radar using the subgroup model. A detailed explanatory caption is given in Fig. 5.

In the MAARSY-15ch configuration, half of the channels have vastly different antenna gain patterns. This fact makes the
configuration better when the subgroup model is applied. We do not present any MC simulations for the phase center model of
the MAARSY-15ch configuration but focus on the subgroup model.

The ambiguity analysis for the MAARSY-15ch subgroup model is illustrated in Fig. 21. There are still a significant amount
of low-distance ambiguities using this configuration, but the distribution and severity of these ambiguities are favorable enough
to expect interferometric capabilities for a majority of all meteor head echo events.

A significant complication was discovered regarding the application of the MUSIC algorithm on the MAARSY-15ch sub-
group model: the vast differences in gain between channels narrows down the peaks in the MUSIC spectrum significantly.
This is usually a desirable property as it allows more precise DOA determination. However, if the narrowing is extreme, a
simple grid search for a peak will become unreasonable costly in terms of computations. As the difference in gain between
the channels increase with zenith angle, the narrowing is a function of elevation, thus making low elevation sources harder to
determine with grid methods.





A peak at 45° elevation from a 10 dB SNR echo would have a peak-width that requires a $3 \times 10^5$ by $3 \times 10^5$ point grid (i.e. $9 \times 10^{10}$ points) to be robustly discovered, as opposed to the 200 by 200 point grid we have used for the MU radar.

To solve this problem we applied the multiple gradient ascent method described in Sect. 3. Instead of only using the maximum point from the grid search as a start value for the gradient ascent, we used the $N$ largest values that were separated from each other by at least $\delta X$ in $k_x, k_y$ space. These $N$ start positions were then explored by a gradient ascent and the largest peak was chosen among these seeds. This proved to be successful for solving the narrow-peak problem of MAARSY with a $N = 50$ and $\delta X = 0.1$ for all tested cases.

The most prominent problem for DOA determination of echoes in the zenith with the MAARSY-15ch configuration is that they are ambiguous with many DOAs below 57° elevation. This is illustrated in the middle upper panel of Fig. 21. If one can restrict the DOA to above 57° elevation, a DOA determination algorithm should be fairly robust at correctly identify the correct direction. This is also supported by the MC simulations of this configuration. The summary results for these simulations are illustrated in Figs. 22, 23 and 24 for $\mathbf{k}_I$, $\mathbf{k}_{II}$ and $\mathbf{k}_{III}$ respectively.

More specifically, the two Input cases in Fig. 22 illustrates that DOA determination is possible above 20 dB and 30 dB for Inputs 1 and 2, respectively, even for sources located outside the radar main lobe. For a source located at the zenith, as illustrated in Fig. 23, much higher SNR is needed for robust DOA determination. The reason for the high algorithm failure probability shown in Fig. 23 is that the bumpy "valley" that constitutes a ring in the sensor response distance, as illustrated in the upper middle panel of Fig. 21, causes the DOA determination to find a local maximum on this ring, which is not included among the output location probabilities.

However, we also simulated a case where an elevation restriction was added to the DOA determination algorithm. It was found that if the algorithm could be restricted to only accept matches above 70° elevation by some reasonable arguments or using a priori data, DOA determination would be stable above 15 dB SNR.

For $\mathbf{k}_{III}$, as illustrated in Fig. 24, the DOA determination suffers the same problem as the MU radar using the subgroup model: there are perfect ambiguities at low elevations. The general DOA determination performance is worse for the MAARSY-15ch configuration then the MU radar, but they still display very similar behaviour for $\mathbf{k}_{III}$. The actual distance for the ambiguity in the upper right panel of Fig. 21 is $d = 0.0002$, i.e 22 times smaller than for the MU radar. It is unrealistic to expect this ambiguity to be resolved within any reasonable SNR for low elevation DOAs.

## 5.4 PANSY radar

The PANSY radar is a special and interesting case when it comes to DOA determination. Firstly, all antennas are distributed in the Z-axis ranging between -2 and +8 meters, the distribution is asymmetric also within the subgroups themselves. Secondly, as is illustrated in Fig. 3, the radar is split into 5 larger collections of subgroups. These disjoint collections are of different sizes and shapes and located relatively far apart. This radar configuration would not be analysable with conventional ambiguity analysis methods due to its complexity. These reasons also make the subgroup gain patterns even more important, a phase center approximation would be outright unphysical to consider. Thus we do not present any results for the phase center model except an ambiguity analysis of $\mathbf{k}_I$, which is illustrated in Fig. 25. Noteworthy in the sensor response distance map is how



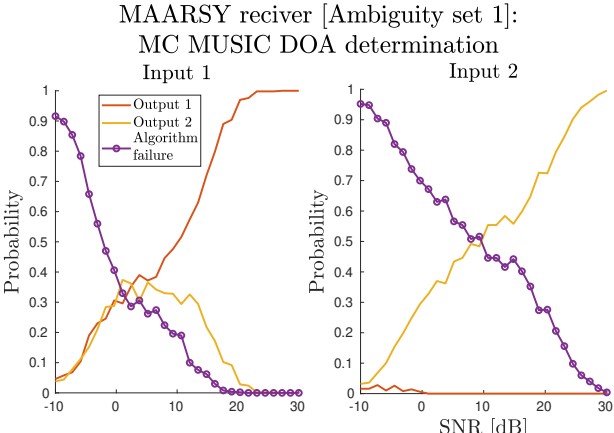

**Figure 22.** The discretised output DOA distribution as a function of SNR and input DOA for the MAARSY-15ch radar. The simulations were made with respect to the ambiguity sets calculated for $\mathbf{k}_I$ using the subgroup model. A detailed explanatory caption is given in Fig. 7.

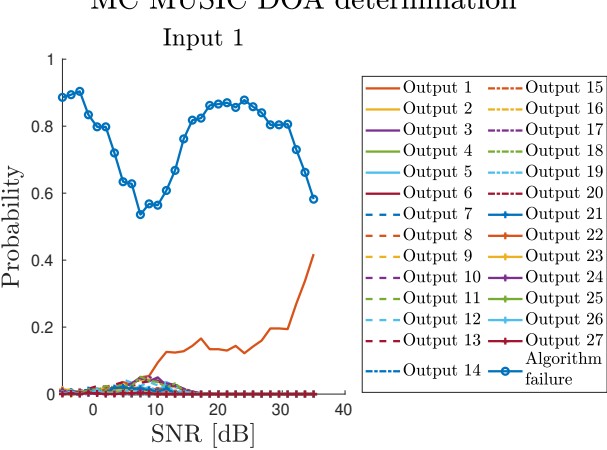

**Figure 23.** The discretised output DOA distribution as a function of SNR and input DOA for the MAARSY-15ch radar. The simulations were made with respect to the ambiguity sets calculated for $\mathbf{k}_{II}$ using the subgroup model. A detailed explanatory caption is given in Fig. 7.

"bumpy" the surface is. This indicates that there are many local maxima and minima that any gradient ascent method could get stuck on.

     Examining the ambiguity results for the subgroup model illustrated in Fig. 26 for $\mathbf{k}_I$, $\mathbf{k}_{II}$ and $\mathbf{k}_{III}$ shows that the "bumpiness" is preserved. Although the inclusion of the subgroup gain rules out many of the previously low-distance ambiguities, a few of them remain at similar distances as those of the phase center model shown in Fig. 25. This is a similar behaviour as for

the MU radar, where many of the phase center model ambiguities are resolved when utilising the asymmetry of the subgroups,





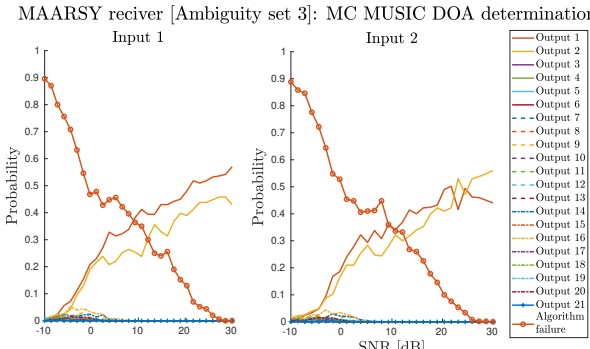

**Figure 24.** The discretised output DOA distribution as a function of SNR and input DOA for the MAARSY-15ch radar. The simulations were made with respect to the ambiguity sets calculated for $\mathbf{k}_{III}$ using the subgroup model. A detailed explanatory caption is given in Fig. 7.

and a few may remain at the same distance. We can also see a potential problem with DOA determination algorithms due to the "bumpiness". In our discretisation process, as outlined in Sect. 2, we need to choose an inclusion distance. The current distance includes all the local peaks illustrated in the example given in Fig. 29. Thus, if one of the local peaks is found by the algorith, the discretising process is not affected. This example contains the MUSIC response of a single simulated meteor head echo.

We have performed several meteor head echo observation campaigns with the PANSY radar system and intend to create an analysis pipeline for the radar system. During that future work, the results presented here and the experience of implementing the DOA determination algorithm on simulated raw data for the PANSY system will be valuable.

     The MC DOA determination simulation summaries are illustrated in Figs. 27 and 28 for $\mathbf{k}_I$ and $\mathbf{k}_{II}$. The results for $\mathbf{k}_{III}$ are practically identical to the results for $\mathbf{k}_I$ but shifted in SNR space. The corresponding Input-Output relation maps are illustrated

in the bottom row of Fig. 26. Examining these summary results shows that there is no need to perform the Bayesian analysis. Ambiguities are not prevalent enough and usually only one clustering will form in low SNR conditions. Instead of ambiguities affecting the quality of DOA determinations, sensor response model errors appear to be most problematic for PANSY meteor head echo observations.

### 5.5    PANSY meteor radar

We only present summary results for the PANSY $2.5\lambda$ meteor receiver system as the very similar standard Jones $2.5\lambda$ system results have been covered in Sect. 5.1.

     For $\mathbf{k}_I$, $\mathbf{k}_{II}$ and $\mathbf{k}_{III}$ the resulting DOA sets $\Omega_X$ and $\Omega_Y$ are illustrated in Fig. 30 together with the Input-Output index maps. Since the antennas are placed at the appropriate distances for a Jones $2.5\lambda$ radar in the x-y plane, but displaced in the z-direction, the actual distances between the antennas are slightly larger then a standard Jones $2.5\lambda$ system. The ambiguity

analysis results in Fig. 30 shows that this has given a slightly negative impact on the severity of ambiguities, when compared to the distances illustrated in Fig. 5. The smallest distance has decreased from $d = 0.43$ to $d = 0.32$.

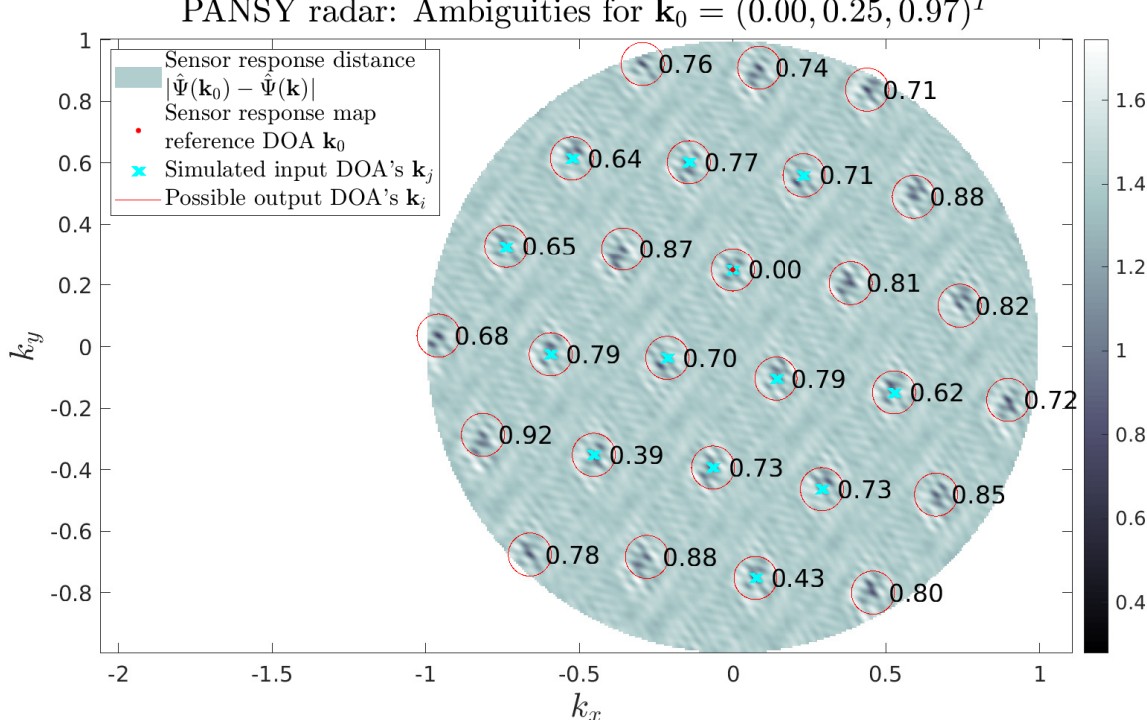

**Figure 25.** Ambiguity analysis summary illustration for the PANSY phase center model. The ambiguity analysis represent the source DOA $\mathbf{k}_I$ as defined in the beginning of Sect. 5. The illustration consists of a sensor response distance $d(\mathbf{k})$ map calculated using Eq. 6. Overlaid on the map are markings according to the legend: the reference DOA $\mathbf{k}_0$, the simulated input DOA set $\mathbf{k}_j \in \Omega_X$ and the possible output DOA set $\mathbf{k}_i \in \Omega_Y$. The sets are calculated as described in Sect. 2.5.

The MC simulations for this system is summarized in Fig. 31 for $\mathbf{k}_I$. Again, the DOA determination dynamics are very similar for $\mathbf{k}_{II}$ and $\mathbf{k}_{III}$, only shifted in SNR space. The magnitudes of these shifts are equal to the differences of the SNR limits given in Table 2. The results are very similar to the standard Jones $2.5\lambda$ system, as expected. One exception is illustrated
in the panel labeled Input 7 of Fig. 31. However, this input DOA is located at an unreasonably low elevation (cf. the bottom left panel in Fig 30) and can be disregarded.

The minimum distance difference between the two systems, $d = 0.43$ for the standard Jones $2.5\lambda$ radar and $d = 0.32$ for the PANSY receiver, should not impact the system performance significantly. For example, comparing Figs. 7 and 31, the algorithm stability threshold for Input 4 is located at 12 dB and 10 dB SNR, respectively. Comparing $\mathbf{k}_{II}$, there is no difference, both
systems give completely stable results above 23 dB SNR.

We have also calculated a simulated Bayesian inference for comparison with the standard Jones system. The minimum needed SNR as a function of observation count is illustrated in Fig. 32 for $\mathbf{k}_I$, $\mathbf{k}_{II}$ and $\mathbf{k}_{III}$, respectively. This analysis shows the impact of the height distribution of the antennas, where $\mathbf{k}_I$ experiences a slight improvement of around 1-2 dB SNR while



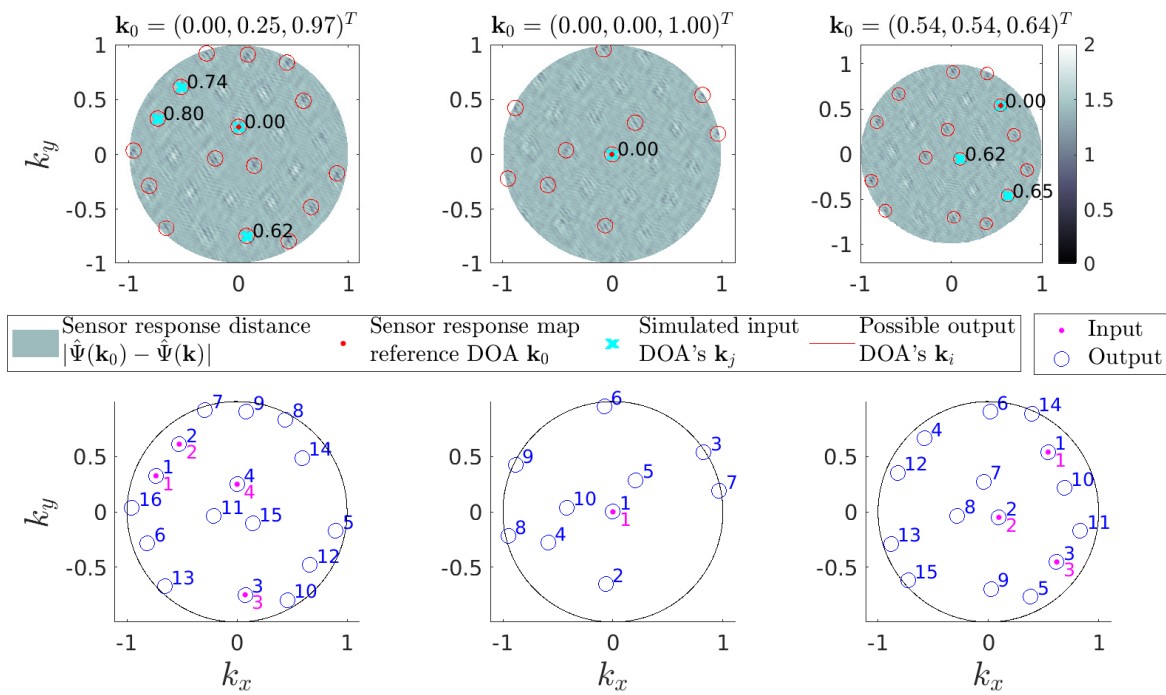

**Figure 26.** Ambiguity analysis summary illustration for the PANSY radar using the subgroup model. A detailed explanatory caption is given in Fig. 5.

$\mathbf{k}_{III}$ is worsened by approximately 4 dB SNR. The latter is consistent with the SNR shift of approximately 5 dB found when

comparing the DOA determination dynamics for $\mathbf{k}_{III}$ with the standard Jones $2.5\lambda$ configuration.

## 6    Conclusions

The main purpose of the ambiguity analysis and the MC DOA determination simulations was to provide improved understanding of DOA determination dynamics and a simulated theoretical reference that is useful when analysing real measurement data.

It is important to note that the results presented here do not take meteor populations into account. We have simply chosen true directions (designated as input) for the simulations, regardless of their probability with respect to some population model. This was done on purpose to prevent disregarding the possibilities of new discoveries. For example, the existence of high altitude radar meteors is still an open question (Gao and Mathews, 2015; Kero et al., 2019). This question could be further addressed using these methods. A common assumption within meteor head echo studies is that the majority of events occur in

the main-lobe (e.g Gao and Mathews, 2015; Schult et al., 2013). Even though this is true, when meteor databases grow larger





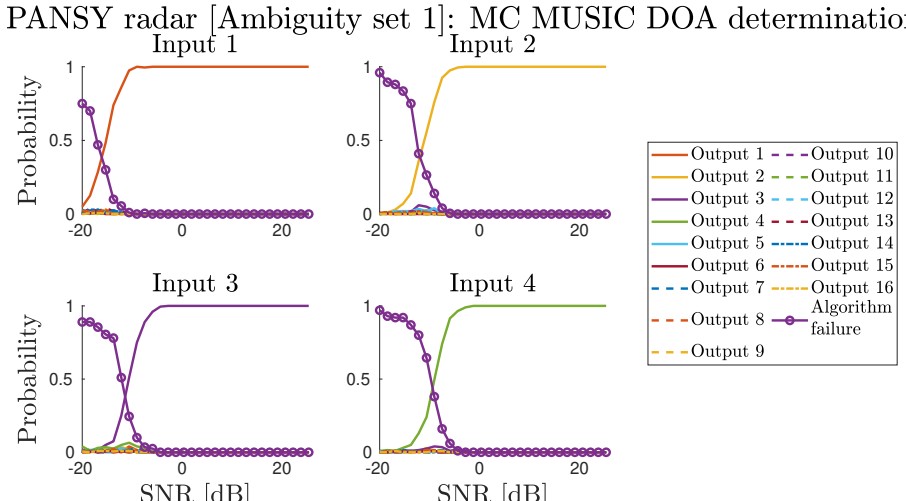

**Figure 27.** The discretised output DOA distribution as a function of SNR and input DOA for the PANSY radar. The simulations were made with respect to the ambiguity sets calculated for $\mathbf{k}_I$ using the subgroup model. A detailed explanatory caption is given in Fig. 7.

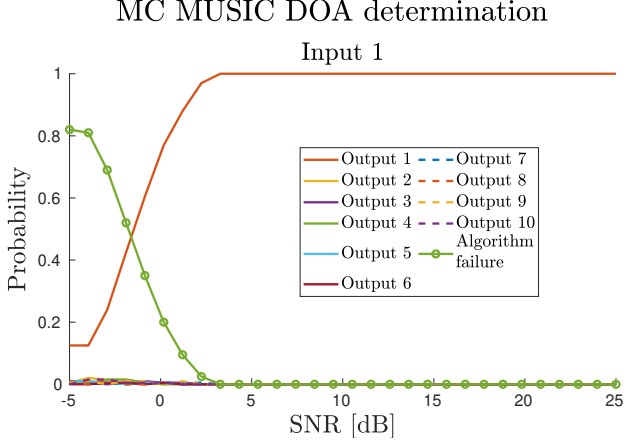

**Figure 28.** The discretised output DOA distribution as a function of SNR and input DOA for the PANSY radar. The simulations were made with respect to the ambiguity sets calculated for $\mathbf{k}_{II}$ using the subgroup model. A detailed explanatory caption is given in Fig. 7.

so should the portion of detections in the side-lobes. For example, Vierinen et al. (2014) showed that high altitude meteor head echoes reported to have been observed with the monostatic EISCAT VHF radar were more likely miss-interpreted side-lobe detections. With respect to other hard targets such as satellites, Vierinen et al. (2019) demonstrated that a majority of such detection using the monostatic EISCAT radars are made in the side-lobes.



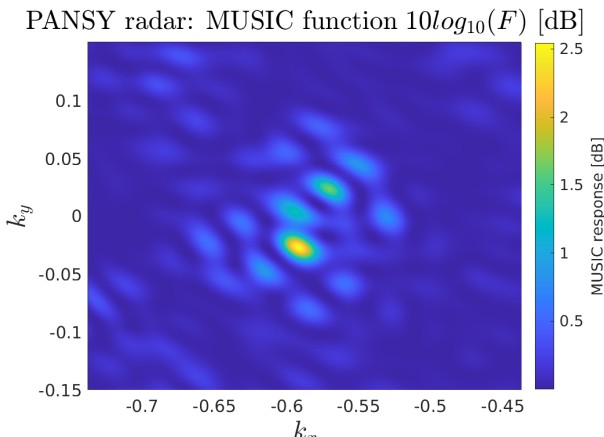

**Figure 29.** Close-up view of the MUSIC function evaluated for a dense grid around a known peak. The MUSIC function was calculated using noisy simulated PANSY subgroup model data at an SNR of 10 dB. The multi-modal nature of the MUSIC function and the widths of the peaks illustrates the difficulty in using a sparse grid to find the peak location without any a priori knowledge.

Using interferometric radar systems to perform meteor head echo measurements, the trajectory can be directly determined (e.g. Kero et al., 2012b). Techniques like the ones presented here can be employed to avoid miss-classifying the DOA, and thereby avoid nonphysical results that are then used in consequent research.

One of the relevant results in this paper is the comparison between using phase center models or subgroup models for the MU, PANSY and MAARSY radars. For the MU radar, even though the subgroup model has ambiguities at low elevations,
this is the expected behaviour in real data as well. Using a simplified model that well approximates the real behaviour close to the zenith is not improving the situation as it may introduce erroneous trajectories in databases. Instead, the known fact of a close to perfect ambiguity can be used as an indicator: if one have several measurements that distribute themselves according to these ambiguities the event can be discarded or put into a special ambiguous category. When the signal originates from the zenith, the subgroup model will expose it to fewer ambiguities and remain stable at lower SNR levels.

For the PANSY radar the situation is similar as the phase center model is outright nonphysical and should not be used. In the case of MAARSY, the inclusion of the subgroup model is the only way to have a chance of resolving arbitrary DOAs unambiguously. However, if the DOA search was restricted to $> 70°$ elevation, stable DOA determination occurs above 15 dB SNR.

These simulations also provided insight into the construction of DOA determination algorithms, as was shown for the
MAARSY, MU and PANSY systems where an additional step had to be implemented due to the topology of the MUSIC function. The success of this method suggests that there may be other optimization algorithms that could further improve performance, such as the Bird Swarm Algorithm (Meng et al., 2016).





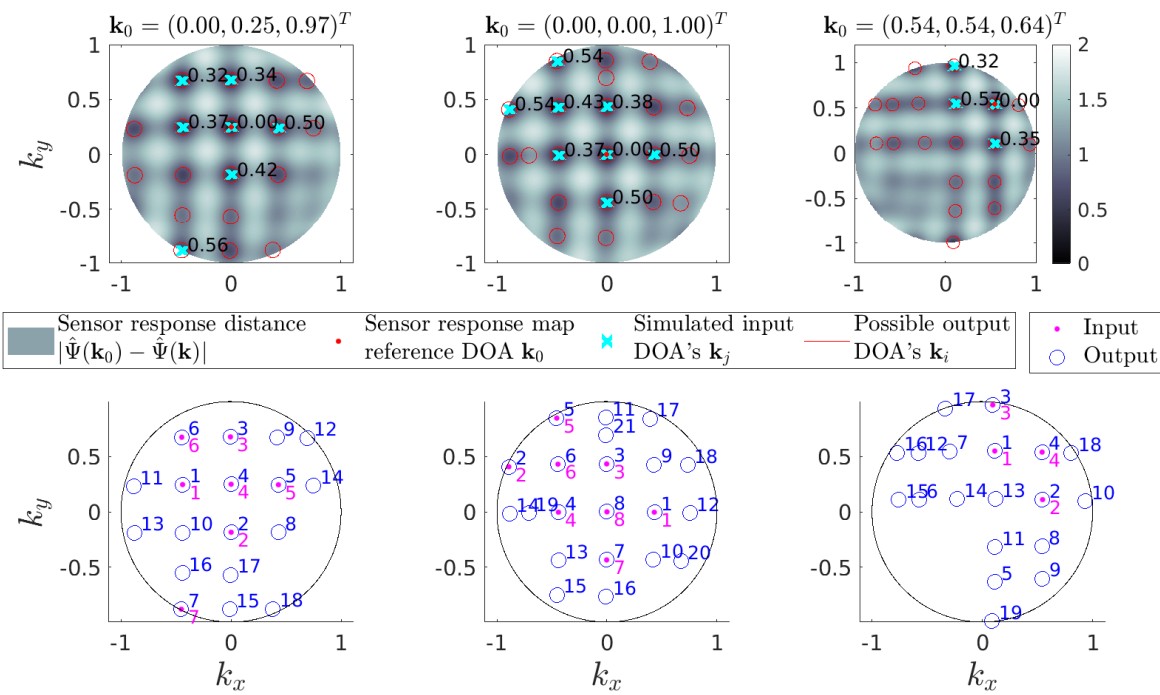

**Figure 30.** Ambiguity analysis summary illustration for the PANSY meteor radar using the phase center model. A detailed explanatory caption is given in Fig. 5.

The comparison between a standard Jones $2.5\lambda$ system and the PANSY meteor radar could help calibrate thresholds for future data analysis pipelines. The comparison showed the advantage of doing ambiguity analysis and MC simulation prior to construction of such pipelines as it can reveal the expected DOA determination performance of a system.

Considering the application of these methods and results on measurement data they provide a reference, not only for SNR limits, but for model validation. If measurements do not follow the dynamics simulated here, and while assuming the pipeline itself is validated and stable, it would point towards the models not representing reality. This makes such simulations a good validation tool for analysis pipelines. For example, it has been frequently shown that multiple-receiver radar systems are in need of phase calibrations (e.g. Chau et al., 2014). In the case of the results presented here: the dynamics would all be modified to some degree if one would add a constant phase to the receivers in our models. This framework of DOA determination simulation provides one with the possibility to test these matters.

We have explored the option of a Bayesian approach to determine the most probable DOA of a target given several measurements distributed among noise-induced ambiguities. The results indicate that this is a suitable method for providing a quantitative probability for which DOA is correct. Using the Bayesian method it appears possible to analyse echoes down to 4



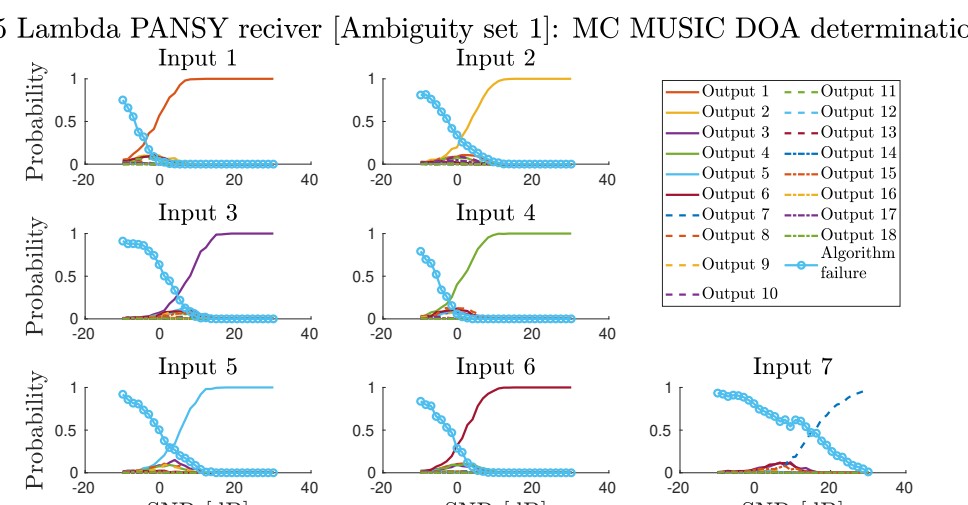

**Figure 31.** The discretised output DOA distribution as a function of SNR and input DOA for the PANSY meteor receiver. The simulations were made with respect to the ambiguity sets calculated for $\mathbf{k}_I$. A detailed explanatory caption is given in Fig. 7.

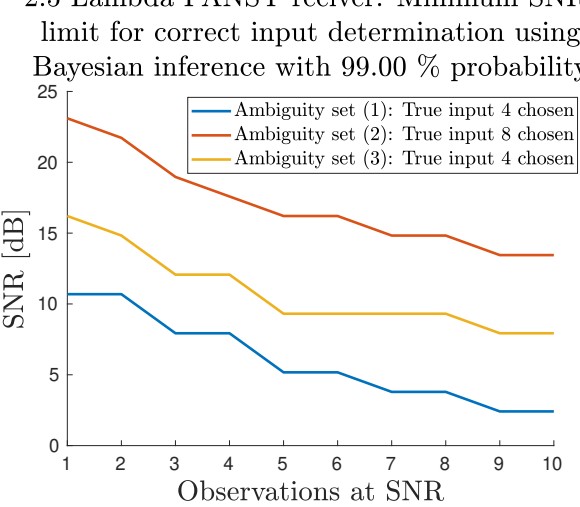

**Figure 32.** Minimum SNR needed as a function of observed outputs in order to correctly identify the true Input direction in 99% of cases using Bayesian inference. A detailed explanatory caption is given in Fig. 9.

dB SNR for both the standard Jones $2.5\lambda$ radar and the PANSY meteor radar, given enough independent data points from the same target.

Lastly, the MC simulations in this paper demonstrated quantitatively that ambiguities are more or less relevant depending on radar system configurations. In systems where ambiguities are not prevalent, the DOA determination failure onset is the



important variable to determine. In systems where noise-induced ambiguities are relevant, it is important to determine the SNR range where they emerge. Our results show that the PANSY system is not affected by noise-induced ambiguities while the MU radar has a small region of SNRs where they could be relevant. The Jones type systems and MAARSY all have relevant noise-induced ambiguities.

Table 2 contains all the MC simulations performed in our study, collected in terms of limiting SNR and input DOA. Any
DOA determination on data with SNR above the limiting value will provide the correct output DOA with $> 99\%$ confidence.

*Author contributions.* DK developed the model code and performed the simulations. DK prepared the manuscript with contributions from JK.

*Competing interests.* No competing interests are present.

*Acknowledgements.* We thank Koji Nishimura for providing the PANSY radar antenna configuration data, Taishi Hashimoto for providing
the PANSY meteor radar antenna configuration data, and Carsten Schult for providing the MAARSY radar antenna configuration data.





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


Atmospheric Measurement Techniques Discussions — Open Access — EGU

**Table 1.** This table contains the $P_{ij}$ probability matrix calculated from a discretisation of the 500 samples MC simulation at around 7 dB SNR illustrated in Fig. 6. The discretisation was made according to the method outlined in Sect. 2.5 and using an inclusion radius of $s = 0.07$. The first 5 output locations, i.e. rows, coincide with the 5 input locations, i.e. columns. Thus, the diagonal in bold denote the probability of MUSIC determining the correct DOA at this SNR level. Each row and column counted from the top-left correspond to the Input-Output index map illustrated in the bottom left axis of Fig. 5. One of these columns can be computed for any SNR. Drawing each row of that column as a function of SNR produces the illustration in Fig. 7. There, each panel corresponds to a column in this matrix. Here "Other" refers to any output DOA not associated with an ambiguity and considered as algorithm failure.

|  | Input DOA | | | | |
|---|---|---|---|---|---|
| | **0.936** | 0 | 0.008 | 0.042 | 0.072 |
| | 0 | **0.338** | 0.098 | 0.036 | 0 |
| | 0 | 0.116 | **0.228** | 0.056 | 0 |
| | 0.012 | 0.098 | 0.096 | **0.624** | 0.038 |
| | 0.030 | 0.004 | 0.006 | 0.062 | **0.710** |
| | 0 | 0.062 | 0.012 | 0 | 0 |
| | 0 | 0 | 0 | 0 | 0 |
| | 0 | 0.096 | 0.022 | 0.088 | 0.040 |
| | 0 | 0.002 | 0 | 0.002 | 0.062 |
| | 0 | 0.078 | 0.064 | 0 | 0 |
| | 0.008 | 0.006 | 0.084 | 0.088 | 0 |
| | 0 | 0.002 | 0.048 | 0 | 0 |
| Output DOA | 0.010 | 0 | 0 | 0 | 0.034 |
| | 0 | 0.064 | 0.002 | 0 | 0 |
| | 0 | 0 | 0 | 0 | 0 |
| | 0.004 | 0 | 0 | 0 | 0 |
| | 0 | 0 | 0 | 0 | 0 |
| | 0 | 0 | 0 | 0 | 0.036 |
| | 0 | 0 | 0 | 0 | 0 |
| | 0 | 0.002 | 0.016 | 0 | 0 |
| | 0 | 0 | 0.002 | 0 | 0 |
| | 0 | 0 | 0 | 0 | 0 |
| | 0 | 0 | 0 | 0 | 0 |
| | 0 | 0.010 | 0 | 0 | 0 |
| | 0 | 0 | 0 | 0 | 0 |
| | 0 | 0 | 0 | 0 | 0 |
| | 0 | 0 | 0 | 0 | 0 |
| | 0 | 0 | 0.022 | 0 | 0 |
| | 0 | 0.002 | 0.010 | 0 | 0 |
| Other | 0 | 0.120 | 0.282 | 0.002 | 0.008 |



**Table 2.** Summary results for all input DOAs used to perform an MC DOA determination simulation. These are the $\Omega_X$ sets for $\mathbf{k}_I, \mathbf{k}_{II}$ and $\mathbf{k}_{III}$. These limits show when the DOA determination with MUSIC for the respective radar system is stable to a level of 99% output probability of the true input. Thus for the given DOA, echoes below the given SNR are possible ambiguous. In the column header Az stands for Azimuth angle and El for Elevation angle in local radar coordinates. An empty SNR indicates that none of the simulations produced a Input-Output correspondence of 99% or higher. For the MU, MAARSY and PANSY radars these results are using the model described by Eq. 30. For the Jones 2.5$\lambda$ and the PANSY 2.5$\lambda$ the model in Eq. 29 was used. The index corresponds to the initial $\mathbf{k}$ vector and the Input-Output maps illustrated for each radar system in Sect. 5.

| Index | Az [°] | El [°] | SNR [dB] | Index | Az [°] | El [°] | SNR [dB] | Index | Az [°] | El [°] | SNR [dB] |
|---|---|---|---|---|---|---|---|---|---|---|---|
| | Jones 2.5$\lambda$ | | | | MU | | | | PANSY 2.5$\lambda$ | | |
| I.1 | -60.25 | 59.70 | 8.97 | I.1 | 26.70 | 30.28 | 3.10 | I.1 | -60.70 | 59.64 | 7.93 |
| I.2 | 60.24 | 59.70 | 15.86 | I.2 | 135.84 | 35.16 | 5.86 | I.2 | 179.89 | 79.34 | 12.07 |
| I.3 | 0 | 46.50 | 17.59 | I.3 | -16.93 | 27.81 | - | I.3 | -0.50 | 47.17 | 17.59 |
| I.4 | 0 | 75.50 | 12.41 | I.4 | -146.41 | 37.47 | - | I.4 | 0 | 75.50 | 10.69 |
| I.5 | 180 | 79.18 | 12.41 | I.5 | 87.16 | 72.13 | -1.03 | I.5 | 60.27 | 60.30 | 14.83 |
| II.1 | 180 | 64.02 | 23.97 | I.6 | 0 | 75.50 | 1.72 | I.6 | -33.53 | 35.73 | 12.07 |
| II.2 | 0 | 90 | 23.97 | I.7 | -161.89 | 81.10 | 1.72 | I.7 | -152.88 | 7.22 | - |
| II.3 | 90 | 64.02 | 23.97 | II.1 | 0 | 90 | 16.67 | II.1 | 90.44 | 64.50 | 23.10 |
| II.4 | -90 | 64.03 | 23.97 | III.1 | -162.24 | 20.67 | 35.59 | II.2 | -65.37 | 11.26 | 23.10 |
| II.5 | 0 | 64.02 | 22.07 | III.2 | 145.18 | 43.81 | 39.66 | II.3 | -0.33 | 64.23 | 21.72 |
| III.1 | 10.84 | 56.53 | 12.41 | III.3 | 45 | 40 | 34.24 | II.4 | -90.46 | 63.85 | 23.10 |
| III.2 | 79.16 | 56.53 | 12.41 | III.4 | 17.27 | 38.06 | 36.95 | II.5 | -28.23 | 15.29 | 21.72 |
| III.3 | 45 | 40 | 12.41 | III.5 | -26.29 | 20.41 | 38.31 | II.6 | -45.84 | 51.65 | 21.72 |
| | MAARSY-15ch | | | III.6 | -138.92 | 25.43 | 36.95 | II.7 | -179.67 | 64.14 | 23.10 |
| I.1 | -139.11 | 67.37 | 23.10 | | PANSY | | | II.8 | 0 | 90 | 23.10 |
| I.2 | 0 | 75.50 | 30 | II.1 | 0 | 90 | 3.28 | III.1 | 11.52 | 55.73 | 16.21 |
| II.1 | 0 | 90 | - | | | | | III.2 | 78.53 | 56.15 | 16.21 |
| III.1 | 45 | 40 | - | | | | | III.3 | 5.55 | 13.09 | 14.83 |
| III.2 | -135 | 31.62 | - | | | | | III.4 | 45 | 40 | 16.21 |
| | DOA > 70° elevation | | | | | | | | | | |
| II.1 | 0 | 90 | 15 | | | | | | | | |