# Peer review of "Probabilistic analysis of ambiguities in radar echo direction of arrival from meteors"

_Atmospheric Measurement Techniques, 2020_

## Referee Comment (RC1) · Anonymous Referee #1 · 20 May 2020

This paper quantitatively examines state-of-the-art methods on evaluating ambiguities in DOA determination using atmospheric radars, and also presents interesting examples of actual applications to several radars using numerical simulations. While the importance of these contributions makes the paper worth publishing in AMT, the present manuscript needs to be substantially revised according to the comments below before it is accepted for publication.

Major comments:

1. This paper aims at two different goals of 1: Understanding the nature of ambiguities in DOA determination of meteor radars, and 2: Providing references to data analysis of several radars. This twofoldness makes this paper very lengthy and thus hard to read. In order to pursue the first goal, which will be of value to most readers who have

general interest on this technical problem, a much fewer examples will suffice, and the number of figures can be easily reduced to half or even less than that of the present manuscript. On the other hand, the detailed analyses of individual radar are useful and important only to a very small group of scientists, who wish to use the named radar for the purpose of meteor observations. Many of the presented figures just confirm trivial results in terms of the first goal. The reviewer thus recommends the authors to essentially rewrite the paper by restricting the goal to be the first one only.

2. Errors in individual antenna characteristics are one of major factors that deteriorate performance of MUSIC algorithm [Ferreol et al., 2006] as indicated by author themselves in Section 2.5, but not examined at all. The assumption made on Line 103 is valid only when all antennas have the same pattern including phase variations with respect to DOA, which is apparently not the case for real radars when the antenna mutual coupling and other factors are considered. Errors in phase calibration is also an important factor that may seriously affect the performance of DOA determination. The authors should include at least some numerical examples representing realistic cases.

3. The reason for selecting the source DOA's of (0,75.5), (0,90), (45,40)degree is not explained. The choice of only considering 0 and 45 degree azimuth planes may be reasonable for Jones radars based on Cartesian grid, but other radars examined here are based on an equilateral triangular grid, which has 60 degree rotational symmetry. Thus 45 degree azimuth will show very similar (if not the same) results as 15 degree azimuth. Authors should choose 30 degree azimuth instead if they try to test most independent cases. They should examine whether these three directions are well representing the nature of all ambiguities before getting into detailed simulations.

Minor comments:

Line 41-42: 'by radar systems' -> 'with interferometry whose base line is longer than half the wavelength'

Figure 3: There are only 7 circles shown for MAARSY radar. A small circle at the center

of the antenna seems to be missing.

Lines 466-470: 'Input 1-5' are defined only in figure captions, but not in the main text. In general, figure captions tend to contain important information not described in the main text. Authors should consider to move those from captions to the main text.

P.38-43: 'Conclusions' should be restricted to summarizing already presented results and ideas, and thus should not introduce new materials and discussions as presented here. It is very difficult to see what is the main conclusion of this paper.

Figure layouts: A new section should not be started before all figures concerning the previous section have been shown. For example, Figs. 22-24 should be presented before Section 5.4 starts, and Figs. 27-32 should be presented before 'Conclusions' starts.

Typographical errors:

Line 20: 'then' -> 'than'

Line 35: 'ecos' -> 'echoes' (two places)

Line 52: 'were' -> 'where'

Line 80: 'then' -> 'than'

Line 406: '19 subarrays' -> 'a single subarray' or '19 elements'

Line 415: 'incoherents' -> 'incoherent'

Line 501: 'the the' -> 'that the'

Line 582: 'trough' -> 'through'

Line 614: 'then' -> 'than'

Line 620: 'then' -> 'than'

Line 648: 'unreasonable' -> 'unreasonably'

Line 674: 'then' -> 'than'

Line 693: 'algorith' -> 'algorithm'

Line 709: 'then' -> 'than'

There are numerous errors regarding 'third-person singular -s'.

---

## Author Comment (AC1) · 1 Jun 2020

Dear Referee #1,

Thank you very much for the useful feedback on the manuscript. As the suggested revisions are quite substantial we believe it is a good idea for us to first present an outline of our planned modifications to the manuscript. This way, if they are satisfactory (to Referee #1) or there have been some misunderstanding, we can mutually confirm the changes before we begin implementation. This is also for the benefit of the other Reviewers, as they can see the plan and discussion.

[Figure]

**Major comments** response:

1. We agree with the assessment. There are several possible actions we can take to fix this problem but we suggest the following:

- Move figures relating to specific radar system results (and include a more complete set) and unnecessary examples to supplementary material only available online

- Move detailed written results / discussion points on specific radar systems that are not necessary to goal #1 to an appendix or remove entirely based on its importance

- As all specific Ambiguity vs SNR probability results will be moved to supplementary material, instead compute and create maps equivalent to Fig. 13 for each radar system and include in the manuscript

- Clarify the remaining written results on specific radar systems with respect to goal #1

- Clarify the conclusions with respect to goal #1

- Perform consequence-revisions based on previous points (e.g. edit description of sections in the introduction)

2. Such simulations have been performed but were not included in the manuscript as they did not affect the Ambiguity dynamics, but rather only the absolute error of unambiguous measurements. These results will now be added to the manuscript as a specific example and with some explanations.

3. When examining Fig. 13 the hexagonal symmetry is illustrated. However, the map is much less "smooth" and "uniform" then one would expect. As such there is no possible

way to choose just a small finite set of source DOA's to represent all possible ambiguity dynamics. However, one might try to choose something in a well determinable region (close to white), something in a indeterminable region (close to black) and something in between. As suggested previously, if we create maps equivalent to Fig. 13 for each radar system we can overlay the selected source DOA's on these maps to motivate the choice of source DOA according to the "low, middle, high" selection. If the source DOA's that we had selected are not representative when doing this, we will change the source DOA and re-run the simulations, otherwise we will keep them as they are and add the motivation to the manuscript.

If these suggested modifications are satisfactory or they need additional re-iteration, please let us know as soon as possible so we can begin the work!

**Minor comments**: We agree with the comments and they will all be addressed in the revised manuscript (except the one concerning Fig. 3: the 7th circle is the one around the entire array).

---

## Referee Comment (RC2) · Anonymous Referee #1 · 2 Jun 2020

The reviewer agrees with the outline of the plan for revising the manuscript proposed by the authors. It will sharpen the focus of the paper, and will make its scientific contribution clearer. While the reviewer does not yet fully understand the idea of the proposed 'maps equivalent to Fig.13', it can be examined after the revised manuscript is completed.

---

## Referee Comment (RC3) · Anonymous Referee #2 · 22 Jun 2020

An array that has a grating lobe with it has a "perfect ambiguity" between the directions of main lobe and the grating lobes. In that case, one cannot distinguish between the true and the ambiguous angle-of-arrivals (AoAs). With random noise added, the true AoA cannot then be discriminated from the directions of grating lobes and side lobes, depending on the SNR. In this manuscript the authors tries to formulate the nature of ambiguity in AoA estimation in mathematical ways together with numerical calculations on the occurrence of ambiguous estimations using the Monte Carlo approaches. Although there are still some significant concerns about the mathematical treatment, the work can be a nice reference for future engineering works of the kind. I recommend it for publication after significant consideration about the issues below.

Major comments: 1. The primary concern is the treatment of the difference in sensor responses, defined in Eq. 6. Consider the case each element response has 180 degree turn from the original. As per the definition, the d(k) exhibit the largest difference. However, a constant phase rotation applied to all elements does not change in direction. This is a fundamental difficulty attached to Eq. 6 as definition of difference in sensor responses with respect to two difference physical directions. In the conventional framework, the difference is defined using inner product. Please elaborate this part and clarify how one can avoid this problem in your framework.

2. Section 2.5 is not clear enough to understand. It is hard to grasp the relations among the variables (k\_i, k\_0, Omega\_x, ...etc). Please consider the following comments. 2-1. It is recommended to illustrate the relations using block diagrams. 2-2. As the authors do not cover the continuous probability distribution but merely discretized distributions based on rather small number of MC trials. As a result, every probability is overestimated because minor events (that do not appear in the limited MC trials) are not involved. Considering this fact, "considered as algorithm failure" (L.203), "If there are no individual..." (L.218-221) becomes incorrect. Did you evaluated the impact of ignoring the probability due to rather limited MC trials? 2-3. L.204: It is unclear that "P\_ij" is joint probability or conditional probability. 2-4. L.208-210: "We then know...Omega(k\_0) is the source.", is it true? It seems "Omega(k\_0)" is the set of ambiguity directions given "k\_0". Isn't it the logic inverted?

3. The mathematical formulation in Section 2.6 looks strange. Usually likelihood function is defined as a function of model parameter x given an observation D; thus L(x|D). Also, in L.253, the sentence "L(D|x) is read as the likelihood of observing D given the parameter x" is strange in the same way. This line talks about PROBABILITY but NOT likelihood. Eq. 18 is probability too. I guess L(x|D) in Eq (17) should be replaced with P(x|D). Please check out and validate them.

Here is a small list of some minor comments. Minor comments: L.14: extra "the" L.101:  $r_i \rightarrow r_j$  L.129: Is "a\_i" a set of wave number vector? Eq (9): please define Psi\_i. Also the difference to Phi should be clarified. L.310: Inconsistent spelling, "maximise" but
"minimize". L.422: What does "instead" mean? Figure 5 and later on: Please add explanations how the intensity scaled [0, 2] as shown in the color bar. L.431: On what reasons the three configurations (I-III) have been chosen? Can you provide some explanations on it?

---

## Author Comment (AC2) · 12 Jul 2020

Dear Referee #2,

Thank you very much for the helpful comments and suggestions! We have below addressed each of the points raised in the review.

Major comments:

1.
To avoid any misunderstanding, what is discussed here is "a constant phase rotation

applied to all elements". In other words, the **same** phase is added to all elements. Thus, such a phase-shift can be extracted from the vector representing radar channels and be written as $\Phi(\mathbf{k})e^{i\theta}$ where $\theta$ is the phase-shift. Such an added phase is practically equal to the temporal component of any received signal. A temporal component is a constant phase added over all spatial locations but that changes in time equally everywhere. As a DOA can be determined from a measured signal at any point in time, such a constant phase should NOT affect the results, which is exactly the case for the definition in Eq. 6. This is also why all (to our knowledge) DOA determination algorithms uses phase-differences between antennas rather then the direct phases as measured by the antennas. The phase difference is used to eliminate this "constant phase" added over all antennas. Therefore, invariance to such a phase is not a problem but rather a desirable property. The conventional framework mentioned that uses the inner product uses a beam-forming approach to DOA determination of a measured signal $\mathbf{x}$ where $|\langle\Phi(\mathbf{k}), \mathbf{x}\rangle|$ is maximized. The inner product will be affected by the addition of a constant phase to all elements as $\langle\Phi(\mathbf{k}_1)e^{i\theta}, \Phi(\mathbf{k}_2)e^{i\theta}\rangle = \langle\Phi(\mathbf{k}_1), \Phi(\mathbf{k}_2)\rangle e^{i2\theta}$. I.e, only by a constant change of phase while the magnitude will **not** change. As Eq. 6, as well as the standard framework, depends on the magnitude, both are invariant with respect to a constant phase-shift of all elements. We realize that the section surrounding the definition of Eq. 6 does not discuss these implied dynamics or alternative implementation enough, thereby giving rise the the concern. To address this, we will add a small discussion along these lines on this topic to that section and clarify the choice we made (to minimize rather then maximize).

2-1.
We agree that the relationships are hard to grasp from only text and the suggestion of a illustrative diagram sounds great! We will add this to the manuscript.

2-2.
We did consider the impact of the limited number of MC samples (due to limited

computational resources) but they were not included in the text. These considerations will now be added to the manuscript to support the results further!

The main considerations are as follows: we can consider the discretization as a set of Bernoulli distributions that defines a success as: "the output DOA fall into the inclusion region" and a failure as "it did not". Then, we can measure the probability parameter $P_i$ for region and Bernoulli distribution $i$ trough the fraction of samples inside that region out of all samples, i.e. $\tilde{P}_i$. This estimators variance (not distribution variance, but variance of the estimator itself) $\mathrm{var}(\tilde{P}_i)$ can be approximated by substituting the distribution variance with the measured Bernoulli variance and applying the central limit theorem $\mathrm{var}(\tilde{P}_i) \approx \frac{\tilde{P}_i(1-\tilde{P}_i)}{N_s}$, where $N_s$ is the number of samples. This estimator variance has a upper limit where the parabola is maximum at $\tilde{P}_i = 0.5$. We used 1000 samples, this makes the largest estimator standard error equal to $\sqrt{(\frac{0.5(1-0.5)}{1000})} = 0.01581....$ In other words, a decent approximation of the upper limit of the estimated probabilities standard error is less then 2%. For the special case where no "successful" observations were made (i.e. no points fall into a region), other considerations can be made to approximate an upper limit for a confidence interval of this probability, but this is still guaranteed to be very small at 1000 samples. Thus, we consider this as a reasonable level of accuracy for the large-scale probabilistic dynamics as a function of SNR that is the goal of this study. The models used will probably account for similar or larger errors if/when our approach is applied on real systems.

Yes, this sentence is incorrect as it should account for fringe possibilities and define the range of scenarios where it applies. It is true for the majority of cases where sampling is sufficient and the dynamics of the DOA determination still form clusters. To illustrate, propose we have a radar system where we have measured a large collection of meteor events, all clearly ambiguous where several distinct clusters of DOA determination outputs are formed for each event. One would then probably try to pick a source DOA $\mathbf{k}_0$ from each of these clusters, generate the corresponding

possible ambiguity sets $\Omega(\mathbf{k}_0)$ and see if any set can contain all clusters. If for NONE of the examined meteors, any set calculated fits the measurements: there is probably something wrong with the model used to generate the sets. We will add a clarification like this to the manuscript in place of the previous erroneous sentence!

2-3.
We will add the traditional Bayes theorem form $P(x|D) = \frac{P(D|x)P(x)}{P(D)}$ to the text again. Originally we intended to make this section easier to follow for someone not familiar with Bayesian probability (from personal experience the $P(D)$ and $P(D|x)$ has proven difficult to explain in text). However, the re-writing seems to have made it more confusing instead. We will re-work this section to use standard notation and be more clear.

2-4.
Unfortunately that sentence is slightly confusing as it is not explained how, with respect to measurements, one could acquire a $\mathbf{k}_0$. When dealing with measurements a $\mathbf{k}_0$ would probably be chosen from the largest clustering of output DOA's, where the SNR is sufficient and a range and perhaps Doppler shift has also been determined that is consistent with the range and Doppler of the other measurements of that event. As such, one want to verify that it is actually a meter echo present in the data and that the chosen sample for $\mathbf{k}_0$ comes from that meteor. If we have such a sample $\mathbf{k}_0$ that we assume comes from the meteor: then yes, the statement is true if we assume that the modeling performed to generate $\Omega$ is representative of the DOA determination behaviour. The true location must be contained in $\Omega(\mathbf{k}_0)$ because only members of this set as an input to the DOA determination algorithm can generate an output at $\mathbf{k}_0$ with high enough probability by definition. That sentence will be replaced with an short discussion on this topic.

3.
Yes, as mentioned, this was an attempt at introducing the concept quickly to someone

without prior statistical knowledge but we realize that is was counterproductive. We will modify this passage to conform to standard statistical nomenclature and put more effort into a seamless introduction of the concept.

Minor comments:

L. 14: Fixed!

L. 101: Fixed in this and several other locations so that $j$ is consistently used as a channel indicator.

L. 129: Yes, it is a set of initial wave vectors. This has been clarified.

Eq. 9: As we are not considering a difference between models for DOA calculation and SNR calculation this should be Phi not Psi! It has been changed everywhere accordingly.

L. 310: Changed all instances of minimize to minimise.

L. 422: Ooops, the word "instead" should not be there! It has been removed.

Figur 5, ect: An explanation on the scale of the color map has been added to the figure captions (it is a direct consequence of the definition of Eq.6, largest length between two points on a $N$-dimensional unit sphere).

L. 431: In our "Author Comment #1" to Reviewer #1 this topic was extensively discussed and we will follow that plan to resolve this question.

---

## Author Comment (AC3) · 13 Jul 2020

All comments need a reply from me before i can click next i think, sorry for the unnecessary notification.

---

## Author Comment (AC4) · 27 Aug 2020

Dear Reviewer,

In my haste before vacation I accidentally miss-wrote in my last review response of major issue 1. Of course the complex space inner product applies a complex conjugation to the first element so that $\langle \mathbf{\Phi}(\mathbf{k}_1)e^{i\theta}, \mathbf{\Phi}(\mathbf{k}_2)e^{i\theta} \rangle = \langle \mathbf{\Phi}(\mathbf{k}_1), \mathbf{\Phi}(\mathbf{k}_2) \rangle e^{-i\theta}e^{i\theta} = \langle \mathbf{\Phi}(\mathbf{k}_1), \mathbf{\Phi}(\mathbf{k}_2) \rangle$, thereby the equivalent definition with the inner product does not need the absolute value to define a distance function $d$ so that anti-directed vectors give maximum separation from the maximum.

I apologize. Best regards, Daniel

---

## Author Response (AR1)

**1 Referee #1 responses**

Thank you very much for the useful feedback on the manuscript.

**Major comments:**

1. We agree with the assessment. We have implemented the following:

- Removed detailed written results, example figures / tables and discussion points on specific radar systems that are not necessary to goal #1

- Computed and created maps equivalent to Fig. 13 for each radar system and included in the manuscript

- Clarified the remaining written results on specific radar systems with respect to goal #1

- Clarified the conclusions with respect to goal #1

- Added two appendices that provide analytical support for choices regarding planar arrays and phase offsets

- Performed consequence-based revisions due to the previous points

2. Such simulations have been performed and some discussions on these results and considerations were added as a subsection called "Impact of phase offsets" and a small appendix called "Impact of known phase offsets on ambiguities". Most of these simulation were assuming measured and known phase offsets. Only two small simulations were performed to test the impact of unknown phase offsets. We believe it is outside the scope of this study to further examine the impact of unknown phase offsets on ambiguities as this would greatly extend the length of the study.

3. We created maps equivalent to Fig. 13 for each radar system to overlay the selected source DOA's on these maps to motivate the choice of source DOA according to the "low, middle, high" selection as well as a wide range of elevations. These considerations were added to the beginning of the results section.

**Minor comments:**

- Line 41-42: 'by radar systems' → 'with interferometry whose base line is longer than half the wavelength' Fixed!

- Figure 3: There are only 7 circles shown for MAARSY radar. A small circle at the center of the antenna seems to be missing. the 8th circle is the one around the entire array!

- Lines 466-470: 'Input 1-5' are defined only in figure captions, but not in the main text. Fixed, noted in the text and defined in a summary table at the end.

- In general, figure captions tend to contain important information not described in the main text. Authors should consider to move those from captions to the main text. We have moved information from several figure captions to the main text.

- P.38-43: 'Conclusions' should be restricted to summarizing already presented results and ideas, and thus should not introduce new materials and discussions as presented here. It is very difficult to see what is the main conclusion of this paper. We have rewritten the conclusions to reflect the results presented previously in the text.

- Figure layouts: A new section should not be started before all figures concerning the previous section have been shown. For example, Figs. 22-24 should be presented before Section 5.4 starts, and Figs. 27-32 should be presented before 'Conclusions' starts. We have moved figure placeholders in the LaTeX file so that they appear earlier in the compiled pdf version of the text. The final placement of figures will be done by the Copernicus copy-writer.

- Line 20: 'then' → 'than' - Fixed

- Line 35: 'ecos' → 'echoes' (two places) - Fixed

- Line 52: 'were' → 'where' - Fixed

- Line 80: 'then' → 'than' - Fixed

- Line 406: '19 subarrays' → 'a single subarray' or '19 elements' - Good catch! Fixed!

- Line 415: 'incoherents' → 'incoherent' - Fixed

- Line 501: 'the the' → 'that the' - Fixed (in multiple places)

- Line 582: 'trough' → 'through' - Fixed

- Line 614: 'then' → 'than' - Fixed

- Line 620: 'then' → 'than' - Fixed

- Line 648: 'unreasonable' → 'unreasonably' - Fixed

- Line 674: 'then' → 'than' - Fixed

- Line 693: 'algorith' → 'algorithm' - Fixed

- Line 709: 'then' → 'than' - Fixed

- There are numerous errors regarding 'third-person singular -s'. Fixed in as many places as we could find

**2   Referee #2 responses**

Thank you very much for the helpful comments and suggestions!

**Major comments:**

1.
To avoid any misunderstanding, what is discussed here is "a constant phase rotation applied to all elements". In other words, the **same** phase is added to all elements. Thus, such a phase-shift can be extracted from the vector representing radar channels and be written as $\mathbf{\Phi}(\mathbf{k})e^{i\theta}$ where $\theta$ is the phase-shift. Such an added phase is practically equal to the temporal component of any received signal. A temporal component is a constant phase added over all spatial locations but that changes in time equally everywhere. As a DOA can be determined from a measured signal at any point in time, such a constant phase should NOT affect the results, which is exactly the case for the definition in Eq. 6. This is also why all (to our knowledge) DOA determination algorithms uses phase-differences between antennas rather then the direct phases as measured by the antennas. The phase difference is used to eliminate this "constant phase" added over all antennas. Therefore, invariance to such a phase is not a problem but rather a desirable property. The conventional framework mentioned that uses the inner product uses a beam-forming approach to DOA determination of a measured signal $\mathbf{x}$ where $\langle \mathbf{\Phi}(\mathbf{k}), \mathbf{x} \rangle$ is maximized. The inner product will be affected by the addition of a constant phase to all elements as $\langle \mathbf{\Phi}(\mathbf{k}_1)e^{i\theta}, \mathbf{\Phi}(\mathbf{k}_2)e^{i\theta} \rangle = \langle \mathbf{\Phi}(\mathbf{k}_1), \mathbf{\Phi}(\mathbf{k}_2) \rangle e^{-i\theta}e^{i\theta} = \langle \mathbf{\Phi}(\mathbf{k}_1), \mathbf{\Phi}(\mathbf{k}_2) \rangle$. I.e, it is invariant to it. We realize that the section surrounding the definition of Eq. 6 does not discuss these implied

dynamics or alternative implementation enough, thereby giving rise the the concern. To address this, we have added a small discussion along these lines on this topic to that section and clarify the choice we made (to minimize rather then maximize).

2-1.
We agree that the relationships are hard to grasp from only text and the suggestion of a illustrative diagram sounds great! We have attempted to simplify the section by adding a flowchart to the manuscript and clarified the explanations in the text.

2-2.
We did consider the impact of the limited number of MC samples (due to limited computational resources) but they were not included in the original text. These considerations have now been added to the manuscript under the subsection "Discretising the problem"!

We have modified the sentence to cover the conditions that are needed for it to be true. We have also added a clarification to the manuscript in place of the previous sentence on the possible checking of erroneous models!

2-3.
We have replaced the old formulation with the traditional Bayes theorem form $P(x|D) = \frac{P(D|x)P(x)}{P(D)}$ and clarified its description. Originally, we intended to make the section easy to follow for someone not familiar with Bayesian probability (from personal experience, the $P(D)$ and $P(D|x)$ has proven difficult to explain in text). Hopefully, it is now resolved.

2-4.
Unfortunately that sentence was slightly confusing. If we have such a sample $\mathbf{k}_0$ that we assume comes from the meteor: then yes, the statement is true if we assume that the modeling performed to generate $\Omega$ is representative of the DOA determination behaviour. The true location must be contained in $\Omega(\mathbf{k}_0)$ because only members of this set as an input to the DOA determination algorithm can generate an output at $\mathbf{k}_0$ with high enough probability by definition. That sentence has be replaced with an short discussion on this topic with disclaimers for when it is applicable.

3.
Yes, as mentioned, this was an attempt at introducing the concept quickly to someone without prior statistical knowledge but we realize that is was counterproductive. We have modified this passage to conform to standard statistical nomenclature and put more effort into a seamless introduction of the concept.

**Minor comments:**

- L. 14: Fixed!

- L. 101: Fixed in this and several other locations so that $j$ is consistently used as a channel indicator.

- L. 129: Yes, it is a set of initial wave vectors. This has been clarified.

- Eq. 9: As we are not considering a difference between models for DOA calculation and SNR calculation this should be Phi not Psi! It has been changed everywhere accordingly.

- L. 310: Changed all instances of minimize to minimise.

- L. 422: Ooops, the word "instead" should not be there! It has been removed.

- Figure 5, etc: An explanation on the scale of the color map has been added to the figure captions (it is a direct consequence of the definition of Eq.6, largest length between two points on a $N$-dimensional unit sphere).

- L. 431: In our "Author Comment #1" to Reviewer #1 this topic was extensively discussed and we have covered these edits in the above review response.

[revised manuscript text omitted]

Minimum search over $\mathbf{k}$

**Set of local minima**
$$\Omega(\mathbf{k}_0) = \{\dots, \mathbf{k}_i, \dots\} = \Omega_X$$

Choose a $\mathbf{k}_0$ using $O_{\mathrm{obs}}$
E.g. mean of largest cluster

Observed ambiguous set
$$O_{\mathrm{obs}} = \{\mathbf{k}_{\mathrm{obs}-l} : l \in [1, N_{\mathrm{obs}}]\}$$

True $\mathbf{k} \in \Omega_X$
but is unknown

**Set of possible outputs**
$$\Omega_Y = \bigcup_{\mathbf{k}_j \in \Omega_X} \Omega(\mathbf{k}_j)$$

Only locations from these sets
can appear as outputs
(excluding algorithm failure)

Given that an element of
$\Omega_X$ is the true $\mathbf{k}$ then
all of $O_{\mathrm{obs}}$ is
contained in $\Omega_Y$
(excluding algorithm failure)

**Figure 3.** Overview of relation between ambiguity sets.
* * *

[revised manuscript text omitted]

---

## Referee Report (RR1)

**Comments to the revised version.**

In the revised edit, the paper has been well refined and now it looks pretty concise. However, the problem of the ambiguity measure defined by Eq. 6 has not been resolved yet. In my first review comments, I tried to make a brief explanation about the problem mentioning the simplest "constant phase rotation" case among those arise by accepting the definition of Eq. 6. The authors answered in part from L. 125 in the paper text, but it does not address the concerns.

In the reply letter, it is mentioned that

As a DOA can be determined from a measured signal at any point in time, such a constant phase should NOT affect the results, which is exactly the case for the definition in Eq. 6.

I do agree with the former half, but I do not understand what you mean by the last part underlined. Applying a constant phase shift to Eq. 6, the resulting distance will look like

$$d = \left| \hat{\Phi}_0 - \hat{\Phi} \cdot e^{j\theta} \right|, \qquad (R1)$$

and this is **not invariant** to the phase shift  $\theta$ .

I try to give another explanation in the following. Think about the following array consisting of *N* antennas. In this case, antenna-O is put to the phase reference to which the phase is fixed to O degree.

The array responses with respect to the vertical (V) and horizontal (H) directions i.e.,

$$\Phi(k_V) = [e^{j0}, e^{j0}, e^{j0}, e^{j0}, e^{j0}, e^{j0}, \cdots] = [1, 1, 1, 1, 1, \cdots],$$
(R2)

$$\Phi(k_H) = [e^{j0}, e^{j\pi}, e^{j\pi}, e^{j\pi}, e^{j\pi}, e^{j\pi}, \cdots] = [1, -1, -1, -1, -1, \cdots].$$
(R3)

When *N* is large, the MUSIC (and most other inner product-based DOA estimation algorithms) spectrum exhibits almost the **maximum** response towards H given V. On the other hand, the "distance" shows the **maximum** (not **minimum** as it is supposed to be in the authors thought) value that is

$$d = \frac{|\Phi(k_V) - \Phi(k_H)|}{N} \sim 2. \tag{R4}$$

This means that the distance measure is not capable of evaluating the ambiguity associated to MUSIC.

As the authors newly mentioned from L. 125, the measure of "ambiguity" can be an authors' choice from numbers of possible definitions. At least, however, the choice must be reasonably compatible with the choice of DOA estimation algorithm (=MUSIC). Simply speaking, the choices of "distance" and MUSIC algorithm are conflicting with each other. In the simulations presented, such conflicts (as in a case depicted above) have been avoided, presumably unintentionally.

In my opinion, this paper is well written having precise descriptions of the methodology and simulation contained. There is the only one contradiction in terms, however; the proposed technique is based on *distance*, while the DOA estimation is based on *inner product*. If my understanding is correct, this problem is too serious to overlook for publishing this paper in this form. Then, there are a few options for the authors can take from.

The most straightforward and seemingly the best option to take is simply to replace the *distance* with *inner product*. Another possible option that I can think of is to add local minimization to replace the *distance* with *the minimum distance by applying an arbitrary but the same phase constant* to all antennas. In an equation form, this can be like,

$$d = \min_{\theta} \left| \hat{\Phi}_0 - \hat{\Phi} \cdot e^{j\theta} \right|. \tag{R5}$$

So far, however, I am not yet pretty sure whether this idea works or not.

Yet another work around is to add discussions in the text about the drawbacks and the limitations arising via the choice of the distance, including; what in principle and how much the two measures (*distance* and *inner product*) are different in a **quantitative way** in association with **use of MUSIC**; in what cases and how much the *distance* can be a good measure as **ambiguity observed with MUSIC**; and, how the authors did and the readers can avoid failures in the proposed technique nevertheless of the two mismatched choices.

---

## Author Response (AR2)

Dear Reviewers,

we really appreciate the time you have taken to explain your intuition and concerns. In these times of covid-19 induced isolation due to cancelled workshops and conferences (such as URSI GA to name one), discussions of ones work and ideas are hard to come by. As exemplified further below, discussion is really important to improve ones work!

We believe to now have fully understood the situation and that there have been two parallel misunderstandings. First of all, the intuition from Reviewer #2 that the inner product is a better indicator for ambiguities than the distance was correct. However, the reason behind the intuition was initially not communicated or maybe not conceptualized enough for us to understand it. The constant phase offset on one element that was discussed is indeed a property of the function $|\langle \mathbf{x}, \mathbf{y} \rangle|$ and not of $|\mathbf{x} - \mathbf{y}|$. The confusion on our side arose since it is not directly this property that makes the inner product a better indicator. It was also unfortunate that in all the specific cases that we had manually validated, this distinction did not matter.

We had considered previously the practical difference between $f(\mathbf{y}) = |\langle \mathbf{x}, \mathbf{y} \rangle|$ versus $f(\mathbf{y}) = \langle \mathbf{x}, \mathbf{y} \rangle$, i.e. that the first possesses less unique information since it does not preserve the relative orientation of $\mathbf{y}$. I.e. if $\mathbf{y}$ is at the same angle from $\mathbf{x}$, but directed "towards" $\mathbf{x}$ or "away from" $\mathbf{x}$, it gives the same value for $f(\mathbf{y})$, but gives very different values when using $\langle \mathbf{x}, \mathbf{y} \rangle$. The distance between two points preserve the relative orientation. This leads us to our second miss-understanding. We had erroneously assumed MUSIC preserves the relative orientation, when indeed it does not. MUSIC is the norm of a projection onto the complement space of the signal subspace, which means that the projection does not have a sense of orientation towards the signal subspace basis vector.

We can now see how the invariance to projection orientation can directly translate to an invariance of a constant phase offset on one of the vectors in the inner product, just like Reviewer #2 stated. In complex vector spaces, the base space is the complex plane $\mathbb{C}$. This means that "projection orientation" is two-dimensional instead of one-dimensional as with real spaces. So, if the projection is a complex number $a = \langle \mathbf{x}, \mathbf{y} \rangle$, any rotation of this number $a e^{i\theta}$ should not affect the result. This can be re-written as $\langle \mathbf{x}, \mathbf{y} e^{i\theta} \rangle$.

To remedy our misunderstandings, we have recalculated every single result in the paper with the new absolute value inner product definition instead of the original distance function. We have rewritten the ambiguity definition part (and as a consequence revised the rest of the manuscript to contain the updated definition), adding also the above discussion, and finally updated the appendices to use the inner product definition (without any need to change their results). We have scrutinized all recalculated results and plots and updated the text in the results section accordingly.

To summarize: there were no large practically important changes in the results. This was expected as nothing changed about the used DOA determination but only the discretization. In terms of the discretization, we have not fine-tuned the limiting values for including ambiguities and second-order ambiguities to match the previous number of ambiguities considered. Therefore, the number of visualised discretizied input/output (and their numbering) have changed in many of the simulations. However, these newly considered locations were unused in almost all situations (P matrix element $\approx 0$ at every SNR) except some cases for the MU and MAARSY radar subgroup models where the input DOA was in an close to undeterminable region (dark color region in Figure 6) and the new discretization had an impact. Comparing old Figure 14, input 2, with new Figure 14, input 8, a previously unconsidered ambiguity now has high probability instead of algorithm failure. The limiting SNR for unambiguous determination is still unchanged, but the onset of algorithm failure is sometimes changed. The onset difference varies from most commonly zero dB to the worst case given in Input 2, Figure 14, where it changed by 20–30 dB.

The difference in discretization can also be seen in Figure 8. But even though this figure looks quantitatively different then previously, qualitatively the dynamics of the resulting P-matrix (Figure 11–12) as a function of SNR is mostly the same. Inputs 5–6 in Figure 9 differ from the previous version, since the old

Figure 9 was calculated without using the scattered gradient ascent version of MUSIC, while this time the updated implementation was used. Additionally, for these two input locations a new ambiguity previously not considered was introduced. The other Inputs are unchanged.

We discovered that a quick way to manually scrutinize the Monte-Carlo data is to animate the DOA Determination as a function of SNR. We have now included such an animation in the data repository (the data upload has not been approved by SND yet, upon request we can send a direct link to the animation).

We will give the manuscript to the Copernicus English copy-editing services to improve grammatical accuracy and overall readability before publication.

[revised manuscript text omitted]

**MU**

| Index | Az [°] | El [°] | SNR [dB] |
|---|---|---|---|
| I.1 |  87.15 |  72.13 |  -1.03 |
| I.2 |  -169.72 |  58.77 |  -6.55 |
| I.3 |  -126.53 |  60.68 | -0.34 |
| I.4 |  3.72 |  52.37 |  -2.41 |
| I.5 |  -146.41 |  37.47 |  - |
| I.6 |  26.70 |  30.27 |  - |
| I.7 |  -33.30 |  56.18 |  -5.17 |
|  I.8 |  0.00 |  75.50 |  -1.03 |
| I.9 | 144.19 | 63.80 | -2.41 |
| I.10 | 109.76 | 49.88 | -6.55 |
|  I.11 |  41.21 |  58.17 |  0.34 |
|  II.1 |  -27.64 |  90.00 |  16.67 |
|  .1 |  45.00 |  40.00 | 34.24 |
|  .2 |  -26.28 |  20.42 |  38.31 |
|  .3 |  145.18 |  43.81 |  39.66 |
|  .4 |  -162.23 |  20.66 |  34.24 |

**PANSY**

| Index | Az [°] | El [°] | SNR [dB] |
|---|---|---|---|
| II.1 |  -98.84 |  90.00 | 3.28 |

**MAARSY-15ch**

| Index | Az [°] | El [°] | SNR [dB] |
|---|---|---|---|
| I.1 | -71.17 | 49.50 | - |
| I.2 | 173.42 | 35.64 | - |
| I.3 | 64.25 | 47.34 | - |
| I.4 | -139.15 | 67.37 | 24.48 |
| I.5 | -111.75 | 19.13 | - |
| I.6 | 18.55 | 33.16 | - |
| I.7 | 102.44 | 13.94 | - |
| I.8 | 0.01 | 75.50 | 30.00 |
| I.9 | 175.81 | 35.15 | - |
| I.10 | -23.32 | 34.67 | - |

**PA** (PANSY 2.5$\lambda$)

| Index | Az [°] |
|---|---|
| I.1 |  -33.55 |
| I.2 |  -60.66 |
| I.3 |  -113.94 |
| I.4 |  179.26 |
| I.5 |  0.00 |
| I.6 |  -0.51 |
| I.7 |  103.93 |
| I.8 | 60.88 |
| I.9 | 145.82 |
| II.1 |  -32.97 |
| II.2 |  -90.28 |
| II.3 |  90.85 |
| II.4 |  179.94 |
| II.5 |  -0.51 |
| II.6 |  -28.23 |
| II.7 |  -45.83 |
| II.8 |  -65.41 |
| II.9 | 119.07 |
| II.10 | 135.85 |
| III.1 |  -45.00 |
| III.2 |  -120.83 |
| III.3 |  -78.60 |
| III.4 |  -169.87 |
| III.5 | -79.52 |
| III.6 | 85.27 |
| III.7 | 5.50 |
| III.8 | 10.80 |
| III.9 | -148.83 |